# Host diet and evolutionary history explain different aspects of gut microbiome diversity among vertebrate clades

Nicholas D. Youngblut[1,6], Georg H. Reischer [2,3,6], William Walters[1], Nathalie Schuster[2], Chris Walzer [4], Gabrielle Stalder[4], Ruth E. Ley[1] & Andreas H. Farnleitner[2,3,5]

Multiple factors modulate microbial community assembly in the vertebrate gut, though studies disagree as to their relative contribution. One cause may be a reliance on captive animals, which can have very different gut microbiomes compared to their wild counterparts. To resolve this disagreement, we analyze a new, large, and highly diverse animal distal gut 16 S rRNA microbiome dataset, which comprises 80% wild animals and includes members of Mammalia, Aves, Reptilia, Amphibia, and Actinopterygii. We decouple the effects of host evolutionary history and diet on gut microbiome diversity and show that each factor modulates different aspects of diversity. Moreover, we resolve particular microbial taxa associated with host phylogeny or diet and show that Mammalia have a stronger signal of cophylogeny. Finally, we find that environmental filtering and microbe-microbe interactions differ among host clades. These findings provide a robust assessment of the processes driving microbial community assembly in the vertebrate intestine.

[1] Department of Microbiome Science, Max Planck Institute for Developmental Biology, Max Planck Ring 5, 72076 Tübingen, Germany. [2] TU Wien, Institute of Chemical, Environmental and Bioscience Engineering, Research Group for Environmental Microbiology and Molecular Diagnostics 166/5/3, Gumpendorfer Straße 1a, 1060 Vienna, Austria. [3] ICC Interuniversity Cooperation Centre Water & Health, 1160 Vienna, Austria. [4] Research Institute of Wildlife Ecology, University of Veterinary Medicine, Vienna 1160, Austria. [5] Research Division Water Quality and Health, Karl Landsteiner University for Health Sciences, 3500 Krems an der Donau, Austria. [6]These authors contributed equally: Nicholas D. Youngblut, Georg H. Reischer. Correspondence and requests for materials should be addressed to N.D.Y. (email: nyoungblut@tuebingen.mpg.de)

Our understanding of the animal intestinal microbiome has now extended far beyond its importance for digestion and energy acquisition, with many recent studies showing that the microbiome contributes to detoxification, immune system development, behavior, postembryonic development, and a number of other factors influencing host physiology, ecology, and evolution[1,2]. Clearly, the adaptive capacity of an animal species is not determined solely by the host genome but must also include the vast genetic repertoire of the microbiome[3]. Concretely understanding how environmental perturbations, host–microbe coevolution, and other factors dictate the microbial diversity in the animal intestine holds importance for the conservation and management of animal populations along with determining their adaptive potential to environmental change[4]. However, we are still far from this understanding, especially regarding nonmammalian species and non-captive species in their natural environment.

A number of factors have been either correlated or experimentally shown to modulate microbiome diversity in the animal intestine[5,6]. While biogeography, sex, reproductive status, and social structure have all been associated with animal gut microbiome diversity in certain animal clades, the consistently dominant drivers appear to be host evolutionary history and diet[7–9]. For instance, diet can rapidly and reproducibly alter the microbiome in humans and mice[10,11]. Still, each individual seems to possess a unique microbiome, and studies on humans and animals have identified microbes whose abundances are determined by host genetics (i.e., heritable microbes)[12,13]. Among animal microbiome studies, the magnitude of these two drivers can differ substantially. For example, diet was the dominant predictor of microbiome diversity in recent studies of great apes[14], mice[15], and myrmecophagous mammals[16]. Other research points to a strong signal of host–microbiome coevolution (i.e., phylosymbiosis) across many animal clades[17,18], and yet other studies have found very little or no effect of host phylogeny (e.g., for chimpanzees or mice)[15,19,20].

A current challenge is determining whether these inter-study discrepancies are the result of technical artifacts inherent to differing experimental designs or whether the modulating effects of host diet and evolution on the gut microbiome do truly differ among host clades and/or microbial lineages. Resolving this question has been hampered by multiple factors. First, most studies have focused on narrow sections of the animal phylogeny (e.g., primates), with a predominant focus on mammals[9]. In fact, the meta-analysis of Colston and Jackson revealed that <10% of studies investigating the gut microbial communities of vertebrates were conducted on non-mammalian species[6]. Although meta-analyses can greatly expand the diversity of hosts analyzed, the heterogeneous sample collection and processing methods employed among individual studies can lead to large batch effects and obscure true biological effects[9,21]. Second, due to the challenge of sample collection and metadata gathering from wild animals, many studies have utilized captive animals. However, the gut microbiome of wild and captive animals can differ substantially[6,22,23], which has led to calls for more studies that assess the microbiomes of wild animals[9,24]. Third, studies vary in how the effects of evolutionary history are assessed. Host phylogenies are inferred from differing molecular data or sometimes only host taxonomy used as a coarse proxy for evolutionary history[6,20,25,26]. Finally, host intra-species variation is often removed (i.e., just one randomly selected sample used per species), or alternatively it is retained but the potential biases and treatment group imbalances are ignored in hypothesis testing[8,26].

To address this challenge, we generate and analyze a very large and highly diverse vertebrate distal gut microbiome 16S rRNA dataset, comprising 80% wild animals that include members of Mammalia, Aves, Reptilia, Amphibia, and Actinopterygii (which diverged from a last common ancestor ~435 MYA). Unlike meta-analyses, this dataset was generated with the same collection methods and molecular techniques performed in the same facility, which reduces batch effects that plague meta-analyses. We utilize a robust analytical framework to resolve the relative importance of host diet and evolutionary history (along with other host characteristics) on gut microbiome diversity. Moreover, we identify particular microbial operational taxonomic units (OTUs) that associate with diet or host phylogeny after controlling for the effect of the other factor. Finally, we utilize eco-phylogenetic methods and co-occurrence analyses to investigate the effects of environmental filtering and microbe–microbe interactions on microbial community assembly in the vertebrate intestine.

## Results

**Sampling strategy.** With the specific aim to cover as much of the breadth of vertebrate hosts animal diversity as possible, we collected fresh fecal samples from the five host classes Mammalia, Aves, Reptilia, Amphibia, and Actinopterygii. Sampling was mostly restricted to animals living in the wild, with some additional samples originating from domesticated livestock and pets (Supplementary Data 1). We generally excluded samples from zoo animals (20 of the 39 samples from captive animals) because artificial habitat, diet, and medication may have strong confounding effects on the natural intestinal communities. No samples were collected from aquariums. The majority of the samples were collected in Central Europe and supplemented with samples from other regions to cover phylogenetic groups lacking extant members in this region (e.g., Afrotheria, Marsupialia, Primates, or Cetacea). To ensure sample origin, samples were gathered by specialized wildlife biologists doing research on the host species in the field. In total, the dataset includes 213 samples from 128 species, each with detailed diet, habitat, and additional metadata (Fig. 1). The number of samples per species varied from 1 to 11 (mean = 1.7), with 50 species having ≥2 samples (Supplementary Fig. 2).

**Low prevalence and limited representation of isolates**. We sequenced the 16S rRNA V4 region from feces or gut contents of all 213 samples and generated OTUs (resolved at 100% sequence identity) with the DADA2[27] pipeline, which produced a total of 30,290 OTUs. Most OTUs (98%) were only detected in ≤5% of samples (Supplementary Fig. 3), which may be due to the high taxonomic and ecological diversity of the hosts. Therefore, we utilized presence–absence for all subsequent OTU-based analyses unless noted otherwise (e.g., for abundance-based beta-diversity metrics). At the phylum level, two clades were found in at least one individual per species: Firmicutes (mainly Clostridia) and Proteobacteria (mainly Betaproteobacteria and Gammaproteobacteria). The next most prevalent phyla were Actinobacteria and Bacteroidetes, which were found in 87% and 86% of host species, respectively (Supplementary Fig. 3).

Mapping phylum-level relative abundances onto the host phylogeny revealed some clustering of microbiome composition by host clade and diet (Fig. 1). Notably, hosts from the same species generally showed similar phylum-level abundances (Supplementary Fig. 2). We quantified this clustering of microbiome composition on the host tree by calculating betadispersion (beta-diversity variance within a group) at each host taxonomic level (class down to species), and indeed we found beta-diversity to be constrained (more clustered) at finer taxonomic resolutions regardless of the beta-diversity metric (Supplementary Fig. 4).

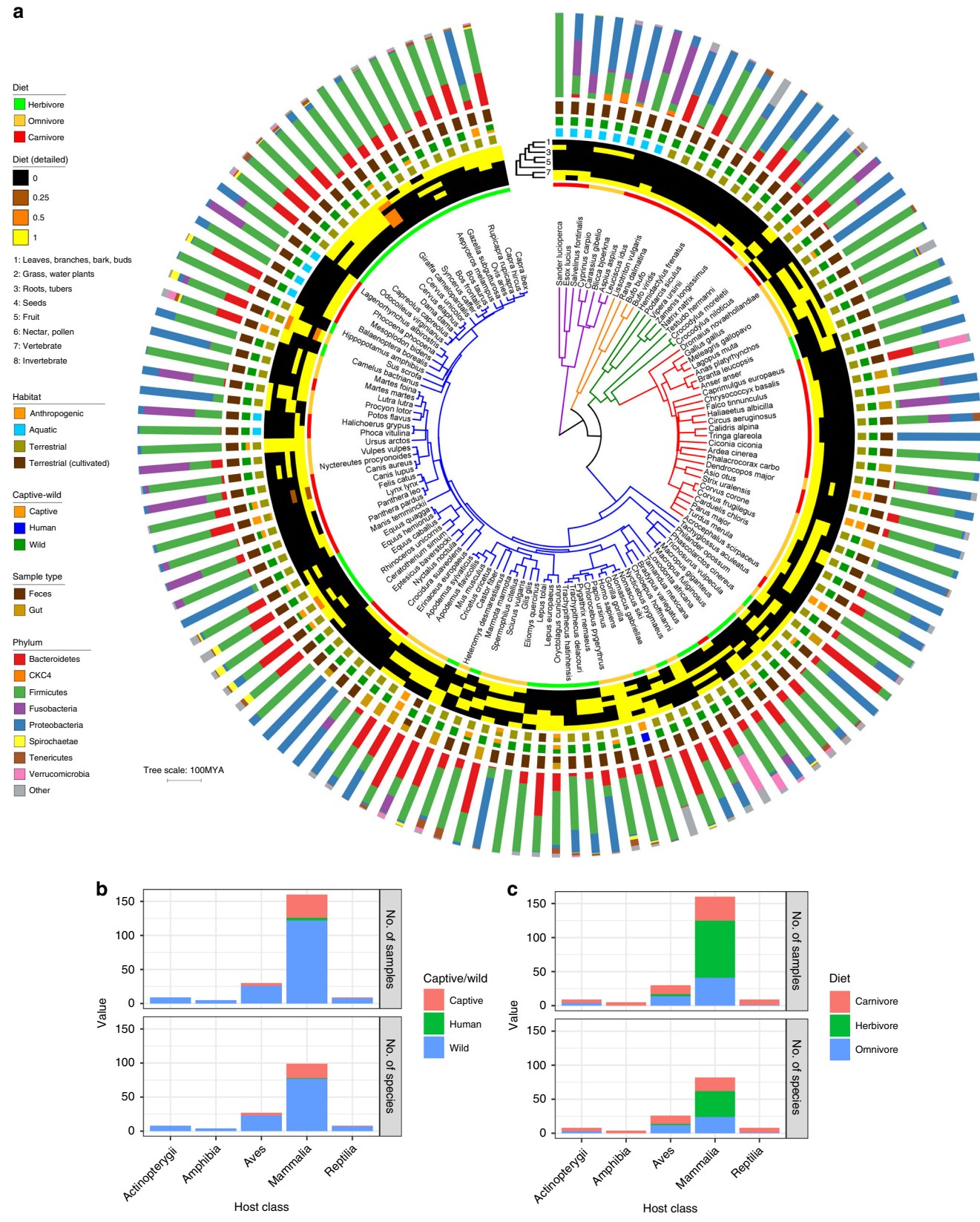

Many of the phylum-level distributions resembled observations from other studies. For instance, Actinopterygii (i.e., ray-finned fishes) samples were mostly dominated by Proteobacteria (Fig. 1), which is consistent with a meta-analysis of fish gut microbiomes[28]. Proteobacteria and Firmicutes were dominant in the Chiroptera species, as seen previously[21]. Fusobacteria abundance ranged from 6% to 35% among the *Crocodylus* species, which is reflective of high Fusobacteria abundance previously

**Fig. 1** Phylum-level grouping of microbiome diversity by host phylogeny and host metadata. **a** The dated host phylogeny was obtained from http://timetree.org, with branches colored by host class (purple = Actinopterygii; orange = Amphibia; green = Reptilia; red = Aves; blue = Mammalia). From inner to outer, the data mapped onto the tree is host diet (general), host diet (detailed breakdown), host habitat, host captive/wild status, the microbiome sample type, and the relative abundances of microbial phyla in each host. Relative abundances are an estimated average generated via subsampling operational taxonomic units from all samples for each host species (subsampling to 5000 for each host species). Note that "Diet (detailed)" information varies among some individuals, and the values shown are averages of the binary yes/no values (no = 0; yes = 1) for each individual. For example, the *Giraffa camelopardalis* samples are from two captive and two wild individuals, so the dietary information somewhat differs, resulting in intermediate values (orange). **b**, **c** show the number of samples or host species per class colored by captive/wild status or diet, respectively. Source data are provided as a Source Data file

observed in alligators[29]. Spirochaete showed high clade specificity for Perissodactyla, Artiodactyla, and Primates, which matches previous observations[30–32]. The CKC4 phylum, which lacks cultured representatives, was markedly abundant in many Actinopterygii samples, reflecting its previous observation in marine species[33,34].

Given the potential for observing novel cultured and uncultured microbes among the phylogenetically diverse and mostly wild hosts, we assessed how many OTUs in the dataset were closely related to cultured and uncultured representatives in the SILVA database. We found that the vast majority (~67%) lacked a BLASTn hit to a cultured representative at a 97% sequence identity cutoff (Supplementary Fig. 5A). Even at a 90% cutoff, ~27% of OTUs lacked a representative. Most OTUs lacking a representative were Bacteroidetes or Firmicutes (46% and 12%, respectively; Supplementary Fig. 5B). Mammalia hosts possessed the majority of OTUs lacking closely related cultured representatives, but still hundreds of OTUs, mainly belonging to Actinobacteria, Proteobacteria, and Verrucomicrobia phyla, were associated with non-mammalian hosts (Supplementary Fig. 5C). In regard to completely novel diversity, ~22% of the OTUs lacked any representative sequence in the entire SILVA r132 database at a 97% sequence ID cutoff. These novel OTUs showed a similar taxonomic composition and distribution among host classes as those OTUs lacking cultured representatives (Supplementary Fig. 5).

Altogether, our assessment of OTU distribution and taxonomy in our dataset revealed that (i) OTUs are sparsely distributed, (ii) host phylogeny constrains beta-diversity, (iii) taxonomic compositions of many host species in our dataset correspond with findings from other studies, and (iv) many OTUs in our dataset, especially those observed in non-mammals, lack cultured representatives.

**Host phylogeny and diet explain microbiome diversity**. We utilized multiple regression on matrices (MRMs) to test how well gut microbiome diversity could be explained by host phylogeny, diet, habitat, geographic location, and technical variation. We chose MRMs because host phylogeny and geographic location can be directly represented as distance matrices (patristic distance and great circle distance, respectively) and measuring host phylogenetic similarity as a continuous variable (patristic distance) versus a discrete variable (taxonomic groupings) alleviates imbalances in representation for specific host taxonomic groups (e.g., Mammalia was highly represented). Host metadata that could not inherently be described as a distance matrix (e.g., the diet components of each species) were converted to distance matrices by various means (see "Methods"). We had no data on the genetic similarity of individuals within host species, and thus we conducted our analysis at the species level. To estimate the effects of intra-species variation in host microbiome and metadata on our MRM analysis, we performed the analysis on 100 subsampled datasets, each comprising one randomly selected sample per

species. Unless noted otherwise, we used this sensitivity analysis approach for all hypothesis testing in this study (see "Methods").

Each of our four MRM models (one per diversity metric) had a significant overall fit ($p < 0.005$ for all MRM models). Host diet and phylogeny were the only significant explanatory variables (Fig. 2). Diet explained a substantial amount of alpha- and beta-diversity variation (~20–30%) and was significant for all diversity metrics tested (i.e., Shannon index, Faith's PD, unweighted Unifrac, and weighted Unifrac). However, host phylogeny was only significant for unweighted Unifrac and explained approximately 15% of the variance. Intra-species variance was lower for weighted versus unweighted Unifrac, so this likely did not cause the lack of association with host phylogeny (Supplementary Fig. 6). Instead, we postulate that host phylogeny mainly dictates community composition but not OTU abundances. Our MRM results were supported by principal component analysis (PCoA) ordinations of weighted and unweighted Unifrac values, which displayed clustering by host taxonomy and diet (Supplementary Fig. 7).

Neither host habitat nor geographic location were significant, likely because these variables were strongly coupled. However, we must note that the experimental design was not directly designed to test this hypothesis (Supplementary Fig. 1). Importantly, the "Technical" covariate, which comprised sample type (feces versus gut contents) and captivity status (wild versus captive) also lacked significance for all models, suggesting no substantial effect of technical variation in our dataset. Also, we did not detect any major outlier samples in our dataset that may be skewing our results (Supplementary Fig. 8). We obtained similar results to our initial MRM analysis when we randomly selected one sample per family instead of per species (Supplementary Fig. 9), which reduced the mammalia:non-mammalia bias from 64% of samples being mammalian to 42%. However, phylogeny was not quite significant (MRM, $p = 0.12$), likely due to the reduced sample size. We found that these results did not substantially change when only including wild animal samples (total samples = 170; total host species = 119) (Supplementary Fig. 10), suggesting that the minor number of captive animals in this study did not substantially contribute to the observed patterns. We no longer observed a significant phylogenetic signal when including just mammals (total samples = 160; total host species = 82), which may be due to the reduced number of host species in the analysis (Supplementary Fig. 11).

We examined whether these patterns change when grouping microbes at coarser taxonomic levels (Supplementary Figs. 12, 13, 14, and 15). The results were mostly consistent with the OTU level; however, weighted Unifrac became significantly associated with host phylogeny, regardless of the taxonomic level. Also, both weighted Unifrac and the Shannon index were no longer significantly associated with host diet at the phylum level.

**Further resolving the effects of host phylogeny and diet**. Our MRM analyses suggest that host phylogeny and diet explain gut microbiome diversity, but this is only one line of evidence, and it

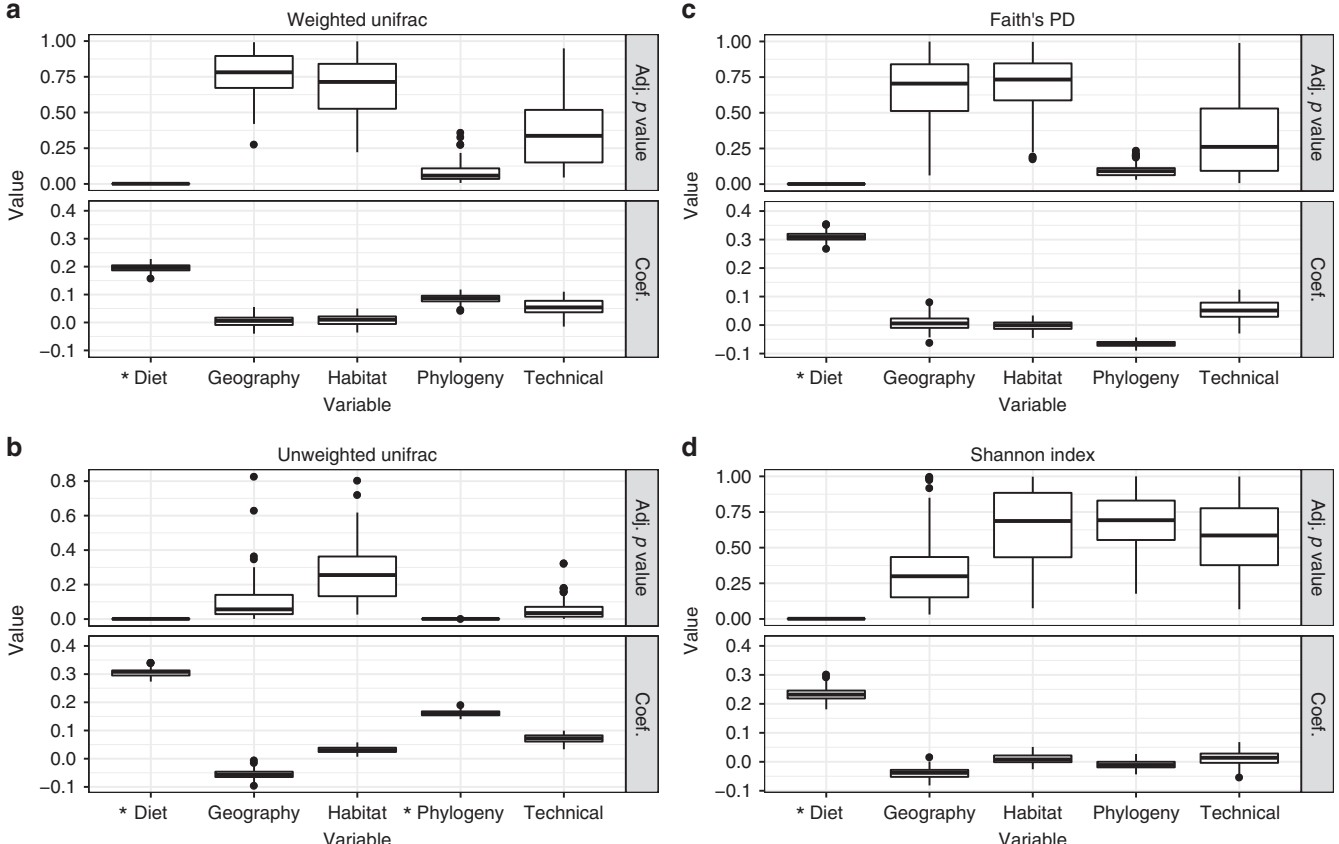

**Fig. 2** Host phylogeny and diet significantly explain the aspects of microbiome diversity. The plots show the BH-adjusted p values (Adj. p value) and partial regression coefficients (Coef.) for multiple regression on matrix (MRM) tests used to determine how much alpha- or beta-diversity variance was explained by host diet, geographic location, habitat, phylogeny, and technical parameters (see "Methods"). The boxplots show the distribution of values obtained when running MRM on each of the 100 random dataset subsets, with each subsample comprising just one sample per species. The boxplots show the MRM rho coefficient and p value for each subsample. See "Methods" for a description of how each distance matrix for the MRM models was generated. Asterisk denotes significance (Adj. $p < 0.05$ for ≥95% of dataset subsets; see "Methods"). Box centerlines, edges, whiskers, and points signify the median, interquartile range (IQR), 1.5× IQR, and >1.5× IQR, respectively. Source data are provided as a Source Data file

does not finely resolve which particular aspects of diversity (e.g., particular OTUs) correspond with host diet and phylogeny. Therefore, we employed complementary tests to our MRM analyses to support and further investigate our findings. While animal host phylogeny is somewhat correlated with diet, our dataset comprised a highly taxonomically diverse set of species with substantially varying diets, which often did not correspond to phylogenetic relatedness (Fig. 1). We exploited this lack of complete correspondence between host phylogeny and diet to decouple the effects of each variable on microbial community diversity.

We used phylogenetic generalized least squares (PGLS) to quantify the association of diet with microbial diversity while accounting for host phylogeny. In support of our MRM results, both alpha- and beta-diversity were significantly explained by host diet (Fig. 3). We also conducted the analysis on individual OTUs and found only 2 OTUs to be significant (Fig. 3c). These OTUs belonged to the Ruminococcaceae and Bacteroidaceae families, respectively. Mapping the distribution of these 2 OTUs onto the host phylogeny revealed that the Ruminococcaceae OTU was associated with many hosts in the herbivorous Artiodactyl clade and also in the southern white-cheeked gibbon (*Nomascus siki*), which is an herbivore in the distantly related primate clade (Supplementary Fig. 16). In contrast, the Bacteroidaceae OTU was predominantly present among multiple distantly related herbivorous clades. The ability of diet to explain overall

community alpha- and beta-diversity but only two OTUs support a hypothesis where diet predominantly selects for functional guilds of microbes (e.g., cellulolytic consortia) rather than specific OTUs.

To assess the effects of host phylogeny while controlling for diet, we utilized tests for phylogenetic signal after regressing out diet. More specifically, we utilized the local indicator of phylogenetic association (LIPA) to assess whether OTU prevalence (i.e., percentage of samples where present) was similar among closely related hosts. We found very little phylogenetic signal of alpha-diversity, which contrasts the substantial association with diet, as observed via the PGLS analysis (Supplementary Fig. 17). This finding is consistent with the MRM analysis results. Also, in contrast to the PGLS analysis, we identified 121 OTUs with significant local phylogenetic signal in the host tree (Fig. 4a). These "LIPA-OTUs" differed greatly in which host clades they were associated with. More specifically, the number of LIPA-OTUs per host species ranged from 1 to 34, with only 21 hosts possessing at least 1 LIPA-OTU. OTU-specific phylogenetic signal was only associated with Mammalia species, suggesting weak or no effects of evolutionary history for non-mammalian hosts. Herbivorous species possessed the majority of LIPA-OTUs, but a minority of these OTUs were associated with some omnivorous and carnivorous species (Fig. 4a). LIPA-OTU composition varied among host clades, regardless of whether they shared the same diet (Fig. 4b), which indicates that the

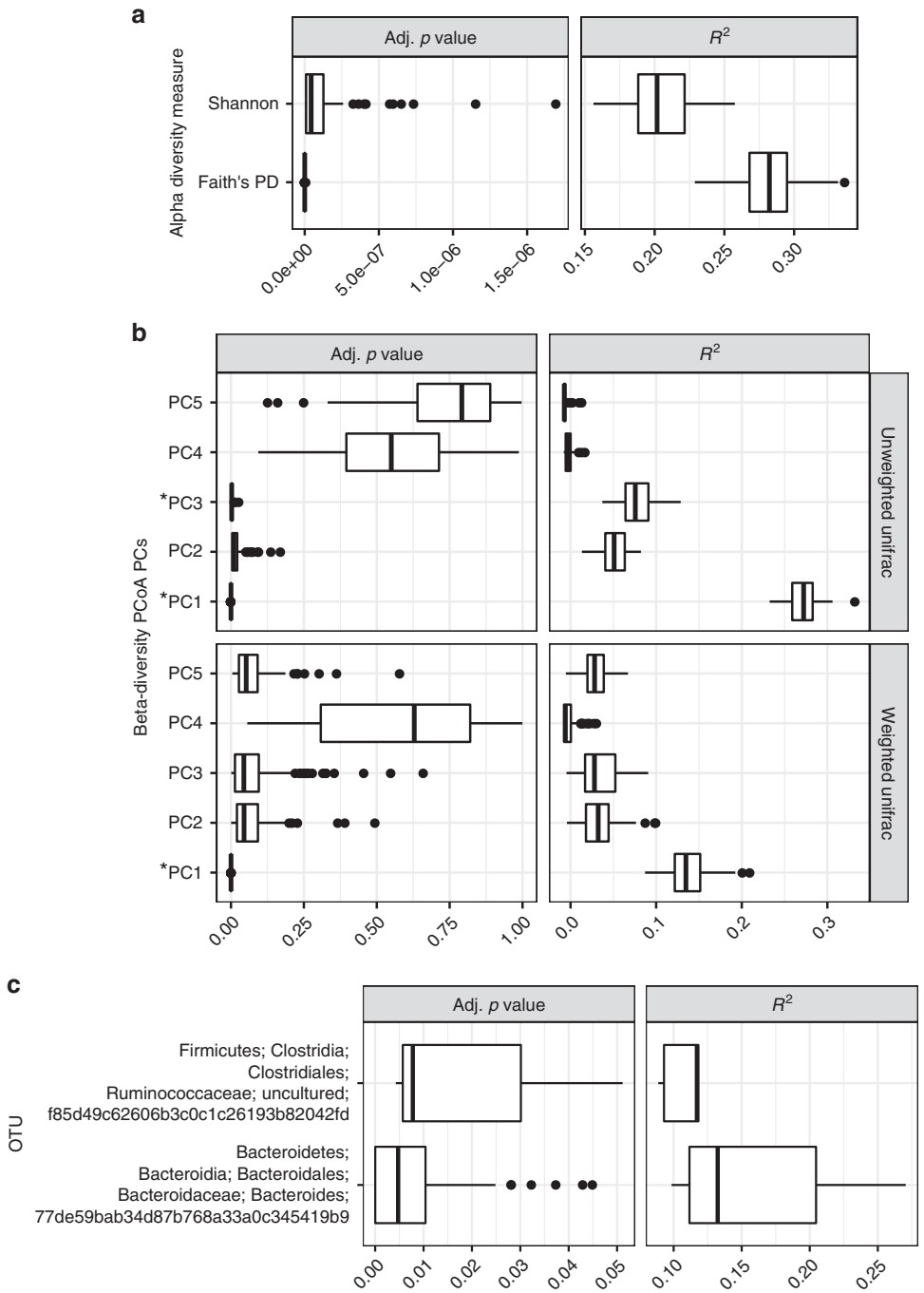

**Fig. 3** After accounting for host phylogeny, diet significantly explained alpha- and beta-diversity components but could only explain the prevalence of two operational taxonomic units (OTUs). The boxplots are distributions of phylogenetic generalized least squares $R^2$ and Adj. $p$ for 100 random subsamples of the datasets (one per species for each subsample). **a** Both alpha-diversity measures were found to be significant. **b** Some principal component (PC) analysis PCs were significantly explained by diet (asterisk denotes Adj. $p < 0.05$). The percentage of variance explained for each unweighted Unifrac PC is 18.1, 6.9, 4.2, 3.6, and 2.1 and each weighted Unifrac PC is 27.2, 10.6, 9.6, 6.4, 6.0, and 5.5. **c** Only two OTUs were found to be significant. Box centerlines, edges, whiskers, and points signify the median, interquartile range (IQR), 1.5× IQR, and >1.5× IQR, respectively. Source data are provided as a Source Data file

phylogenetic signal is indeed a result of host evolutionary history and not contemporary diet. LIPA-OTUs were most predominant among Artiodactyla species, with Primates and Perissodactyla ranked a distant second and third (Fig. 4b). This finding suggests that the effects of host evolutionary history within Mammalia are most pronounced for Artiodactyla. Interestingly, there was no OTU-specific phylogenetic signal for any macropods, even though they are foregut fermenters similar to the Artiodactyla.

The same is true of Carnivora species, except for 2 members of the Felidae clade (*Felis catus* and *Panthera pardus*). Altogether, these findings support the hypothesis that mammalian evolutionary history dictates the prevalence of certain OTUs.

The LIPA-OTUs belonged to seven bacterial phyla and one archaeal phylum (Fig. 4c; Supplementary Fig. 18). Firmicutes was dramatically more represented than other phyla, with Bacteroides the second-most common. Members of Bovidae consistently had

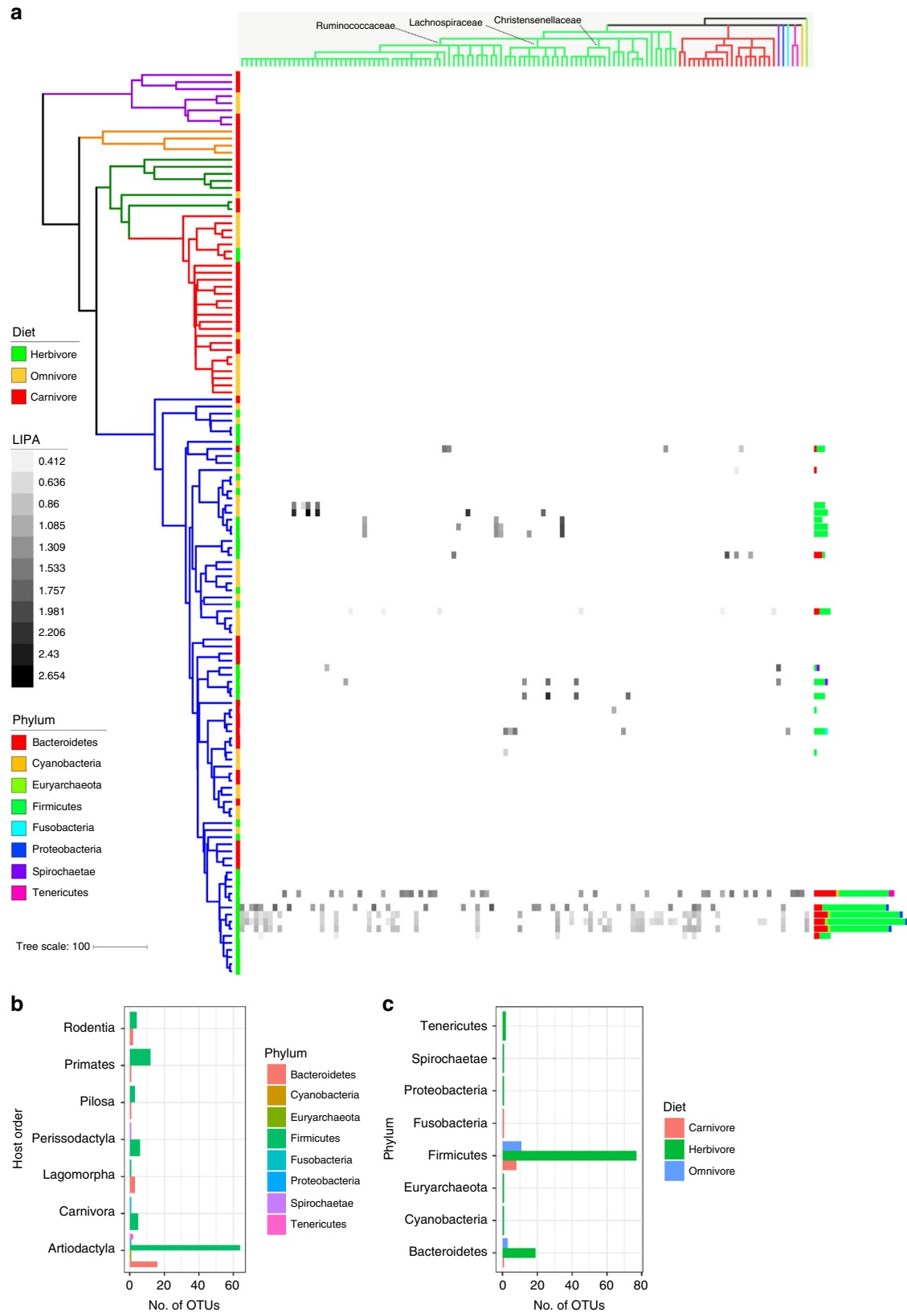

the highest numbers of these two phyla; this finding is supported by Sasson and colleagues[13], who only identified Bacteroides and Firmicutes to be heritable in cattle. The majority of the Firmicutes OTUs were members of the Ruminococcaceae, and while most of Ruminococcaceae OTUs were associated with Artiodactyla hosts, some were also observed in certain members of the Primates,

Rodentia, and Perissodactyla. Other OTU clades with significant phylogenetic signal included the genera *Christensenella*, *Blautia*, and *Methanobrevibacter*, which were all found to be consistently heritable among multiple human cohort studies[12,35]. Interestingly, while humans are represented in this dataset, and a few OTUs were associated with some of the primate species, no OTUs

**Fig. 4** Many operational taxonomic units (OTUs) display a local phylogenetic signal in specific host clades after accounting for diet. **a** The phylogeny is the same as shown in Fig. 1. The heatmap depicts local indicator of phylogenetic association (LIPA) values for each OTU–host association, with higher values indicating a stronger phylogenetic signal of OTU presence (with diet regressed-out). White boxes in the heatmap indicate non-significant LIPA indices. The dendrogram on the top of the heatmap is a cladogram based on the SILVA-derived taxonomy for each OTU (see Supplementary Fig. 18 for the full taxonomy). The dendrogram is colored by phylum. The bar plots in **b** and **c** show the number of OTUs with a significant LIPA index per host (OTUs are colored by phylum; the number of OTUs per host ranges from 1 to 34). **b** The bar plots summarize the number of significant OTUs per host order and diet. The bar plots in **c** are the same as **b** except the data are grouped by OTU phylum. Source data are provided as a Source Data file

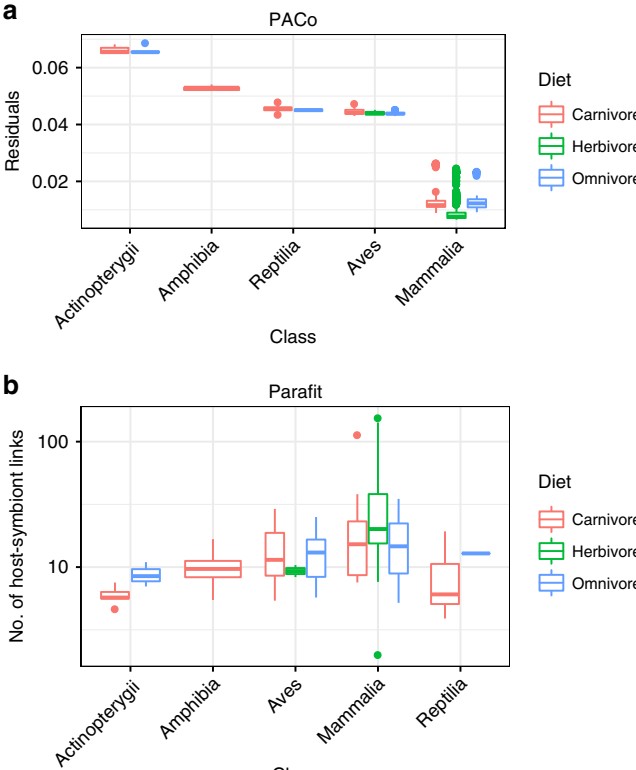

**Fig. 5** Procrustean approach to cophylogeny (PACo) and Parafit show a stronger cophylogeny signal for Mammalia versus non-mammals. **a** Boxplots of PACo residuals between hosts and operational taxonomic units (smaller residuals means a stronger cophylogeny signal), with residuals grouped by host class and diet. **b** Boxplots of significant host–symbiont links as determined by Parafit analysis, with links grouped by host class and diet. For both PACo and Parafit, 1000 permutations were performed on each of the 100 dataset subsets. Box centerlines, edges, whiskers, and points signify the median, interquartile range (IQR), 1.5× IQR, and >1.5× IQR, respectively. Source data are provided as a Source Data file

showed a phylogenetic signal with humans (Fig. 4a). Among some very closely related OTUs, we observed that host clade specificity differed, suggesting that these taxa have diversified via adaptive specialization for particular hosts (Supplementary Data 2; Supplementary Fig. 18).

**A stronger pattern of cophylogeny in Mammalia**. Our finding that only Mammalia possessed OTUs with local phylogenetic signal suggests that the effects of evolutionary history on intestinal microbiome diversity may be stronger for Mammalia versus non-mammalian species. We investigated this finding by performing cophylogeny analyses, which determines whether the phylogenies of the host and symbiont (microbe) correspond in their branching patterns. While a positive correlation can be the

result of other processes besides co-cladogenesis[36], the pattern is consistent with a model of host–symbiont coevolution. We first utilized Procrustean approach to cophylogeny (PACo[37]), which performs Procrustes superimposition to infer the best fit between host and symbiont phylogenies based on symbiont occurrences in the hosts. This permutation-based approach does not rely on distribution assumptions. Moreover, the analysis generates residuals of the Procrustean fit, which describes the contribution of each individual host–symbiont association to the global fit (smaller residuals means a better fit).

The PACo analysis showed a significant global fit, regardless of intra-species heterogeneity (PACo, $p < 0.002$ for all dataset subsets). Host–microbiome residuals decreased in the order of Actinopterygii > Amphibia > Reptilia ≥ Aves > Mammalia, with the most dramatic decrease between Aves and Mammalia (Fig. 5), indicating that Mammalia show the strongest signal of cophylogeny. The residuals significantly differed by both host class and diet (analysis of variance, $p = 1\mathrm{e}{-16}$ for both), but the effect size was much larger for class versus diet ($F$-value of 972.3 versus 536.3). Thus, while diet may somewhat confound the signal of cophylogeny, it is likely not the main driver. Conducting PACo on just mammalian species still showed a significant global fit (PACo, $p < 0.002$), and we found that Artiodactyla have the smallest distribution of residuals (Supplementary Fig. 19A). Excluding all Artiodactyla samples did not substantially change the results (PACo, $p < 0.003$); neither did sub-sampling just one sample per family in order to decrease the imbalance of host species per clade (PACo, $p < 0.003$; Supplementary Fig. 19B, C).

We additionally evaluated patterns of cophylogeny with the Parafit analysis, which is also a permutational method but assesses similarity of principal coordinates derived from the host and symbiont phylogenies. As with PACo, the global Parafit test was significant (Parafit, $p < 0.001$), and Mammalia showed the strongest signal of cophylogeny (Fig. 5). Altogether, these data support a model of host–microbe coevolution, with Mammalia displaying the strongest cophylogeny signal.

The stronger signal of cophylogeny among mammals may be the result of more transient environmental microbes in the guts of non-mammals. We assessed this possibility by mapping taxa to the Earth Microbiome Project (EMP)[38] 16S rRNA dataset (No. of samples: Animal = 317, Human = 206, Sediment = 259, Soil = 193, Water = 242) and using the indicator value analysis[39] (IndVal) to assess the specificity of taxa to (i) the guts of mammals versus non-mammals in our dataset and (ii) biomes in the EMP dataset. We found 32 bacterial genera and 1 archaeal genus to show significant specificity for mammals or non-mammals in our dataset and also significant biome specificity in the EMP dataset (IndVal, Adj. $p < 0.05$; Supplementary Fig. 20A). Moreover, the non-mammal associated taxa had a significantly higher specificity for environmental EMP biomes (Wilcox, $p < 0.006$); Supplementary Fig. 20B), which corroborates our hypothesis. Genera did not contribute equally to this signal, and actually many non-mammal and mammal specific genera were only specific to human and/or animal biomes in the EMP dataset. Still, more non-mammal-specific genera (e.g., *Desulfolobus* and *Hyphomicrobium*) were strongly associated with

environmental biomes relative to mammal-specific genera. The only mammal-specific genus to show a strong environmental association was *Paludibacter* (Bacteroidetes phylum).

**Environment filtering and microbe–microbe interactions**. Our findings that diet and host evolutionary history significantly explain microbiome diversity indicate that environmental filtering plays a substantial role in microbial community assembly. In order to further test this notion and to assess how environmental filtering may differ among host clades, we utilized two ecophylogenetics analyses: mean phylogenetic distance (MPD) and mean nearest taxon distance (MNTD). These tests assess the degree of phylogenetic clustering within each sample (host) relative to a permuted null model. Assuming phylogenetic niche conservatism (i.e., closely related taxa overlap along niche axes), then host diet or gut physiology may select for phylogenetically clustered taxa with overlapping niches, while in the absence of such strong selection, competition via niche conservatism would lead to phylogenetic overdispersion[40]. Phylogenetic overdispersion may also result from facilitation (i.e., beneficial microbe–microbe interactions), such as when distantly related taxa form consortia to break down complex plant polymers[40]. MPD is more sensitive to overall patterns of phylogenetic clustering and evenness, while MNTD is more sensitive to patterns at the tree tips[41].

We found that the majority of host species showed significant clustering for MNTD, with close to half for MPD (Fig. 6). Very few species showed phylogenetic evenness. Of those that did, all belonged to the Artiodactyla, except for the long-eared owl (*Asio otus*; Fig. 6). In support of these findings, Gaulke and colleagues[42] also found lower signals of phylogenetic clustering in the Artiodactyla relative to other mammalian clades. These findings suggest that community assembly differs between Artiodactyla and non-Artiodactyla mammals, with microbe–microbe competition and/or facilitation surpassing gut environmental filtering among Artiodactyla species.

We next tested how microbes co-occur among hosts, which can be influenced by selective pressures or microbe–microbe interactions. Specifically, we conducted a co-occurrence analysis to determine which OTUs significantly positively or negatively co-occurred relative to a permuted null model. Our analysis revealed that almost all significant co-occurrences were positive (Fig. 7a; Supplementary Fig. 21A). The co-occurrence network consisted of four sub-networks, each with differing taxonomic compositions and existence of "hub" OTUs (Fig. 7d). Sub-networks 1 and 2 were dominated by Ruminococcaceae and Peptostreptococcaceae, with Ruminococcaceae OTUs acting as central hubs in both (Supplementary Fig. 22). Sub-network 3 contained an Enterobacteriaceae (Proteobacteria) OTU hub and also possessed more members of Clostridiaceae, Lachnospiraceae, and Enterobacteriaceae. Sub-network 4 did not have a strong hub OTU and contained the most taxonomic diversity (Fig. 7d). Interestingly, *Methanobrevibacter* OTUs were only found in sub-network 1 and significantly co-occurred with *Christensenellaceae* OTUs as previously seen in a large human cohort study[35]. The presence of OTUs from each sub-network differed substantially among host clades (Fig. 7b). Sub-networks 3 and 4 were generally most prevalent in many host orders, with only one of the two networks being highly prevalent. Sub-network 1 was only prevalent in the Artiodactyla, suggesting strong host specificity of this microbial consortium. In support of this finding, the network contained a substantially higher proportion of OTUs with local phylogenetic signal among hosts relative to the other sub-networks (Fig. 7d). Sub-network 2 was only prevalent in four mammalian orders: Artiodactyla, Diprotodontia, Pilosa, and Primates. The sub-networks showed significant distributional shifts among diets

(Kruskal–Wallis, $p < 2.2e-16$; pairwise Wilcox test, Adj. $p < 0.05$ for all tests), with sub-networks 1 and 2 being most prevalent among herbivores, sub-network 4 dominating in omnivores, and sub-networks 3 and 4 showing similar prevalence among carnivores (Fig. 7c).

**Discussion**

While various studies have shown that host diet and phylogeny modulate the animal intestinal microbiome[5,6], we have expanded on this previous work by performing a robust assessment of each factor's effect on a homogeneously generated dataset of highly diverse and predominantly wild animals. Because our dataset consisted of animals from diverse lineages that consume a range of dietary components, we were able to decouple the effects of host phylogeny and diet on both aggregate diversity metrics and at the individual OTU level. We employed multiple analytical methods to support our findings, and we also directly assessed the sensitivity of our analyses to intra-species microbiome and metadata heterogeneity, which has been found to be non-trivial[7,14,43,44]. We did not have inter-individual replicates for some host species in our dataset, which limited our ability to determine the impact of this factor for certain host clades; nevertheless, our findings suggest that host diet and evolution are strong modulators despite the intra-species variability that we measured. We did not observe that habitat or geographic distance explained microbiome diversity, which is consistent with some animal microbiome studies[6,25] but not others[6,21,45]. Possibly, these factors may only modulate the microbiome of certain host clades, or our dataset is underpowered in regard to testing these potential modulators.

Only a couple of very coarsely resolved taxonomic groups were present in (nearly) all host species (Supplementary Fig. 3). This finding suggests that most microbial clades, especially finely resolved clades, are somewhat constrained to certain host clades. Indeed, we did find beta-diversity to be more constrained at finer host taxonomic levels (Supplementary Fig. 4). The largest exception to this trend was the Clostridiales order, which we observed in ~98% of host species (Supplementary Fig. 3). Many members of Clostridiales generate resistant spores, which may allow for high inter-species or environment–host migration. This process could generate source–sink dynamics, where Clostridiales pass through specific gut environments only transiently, but their high migration rates from source hosts, soil, water, etc. continually replenish these ephemeral sink populations. In contrast, our data support true specialization of certain Clostridiales for specific host clades. First, all Clostridiales genera specific to mammals or non-mammals in our dataset were specific to animal/human samples and not environmental samples in the EMP dataset (Supplementary Fig. 20). Second, we found that the majority of OTUs displaying a local phylogenetic signal belonged to Clostridiales (Fig. 4). Importantly, these Clostridiales OTUs showed specificity for differing host clades, which have different exposures to potential source communities, and thus the signal of host specificity is unlikely to have resulted from transient populations maintained by high migrations rates. While only two OTUs were significantly modulated by host diet after controlling for phylogeny, one belonged to Clostridiales (Fig. 3), suggesting that specialization to specific host clades (and in some instances, diet) contributed to adaptive speciation in this lineage.

New culturomics techniques are greatly reducing the number of uncultured microbes in the human gut[46]; however, our analysis suggests that microbes from other animals are far less represented (Supplementary Fig. 5). This even applies to Mammalia, which have received the lion's share of focus for gut microbiome studies[6]. Our limited knowledge of gut-inhabiting microbes of many animals is typified by the CKC4 phylum, which we found to be a

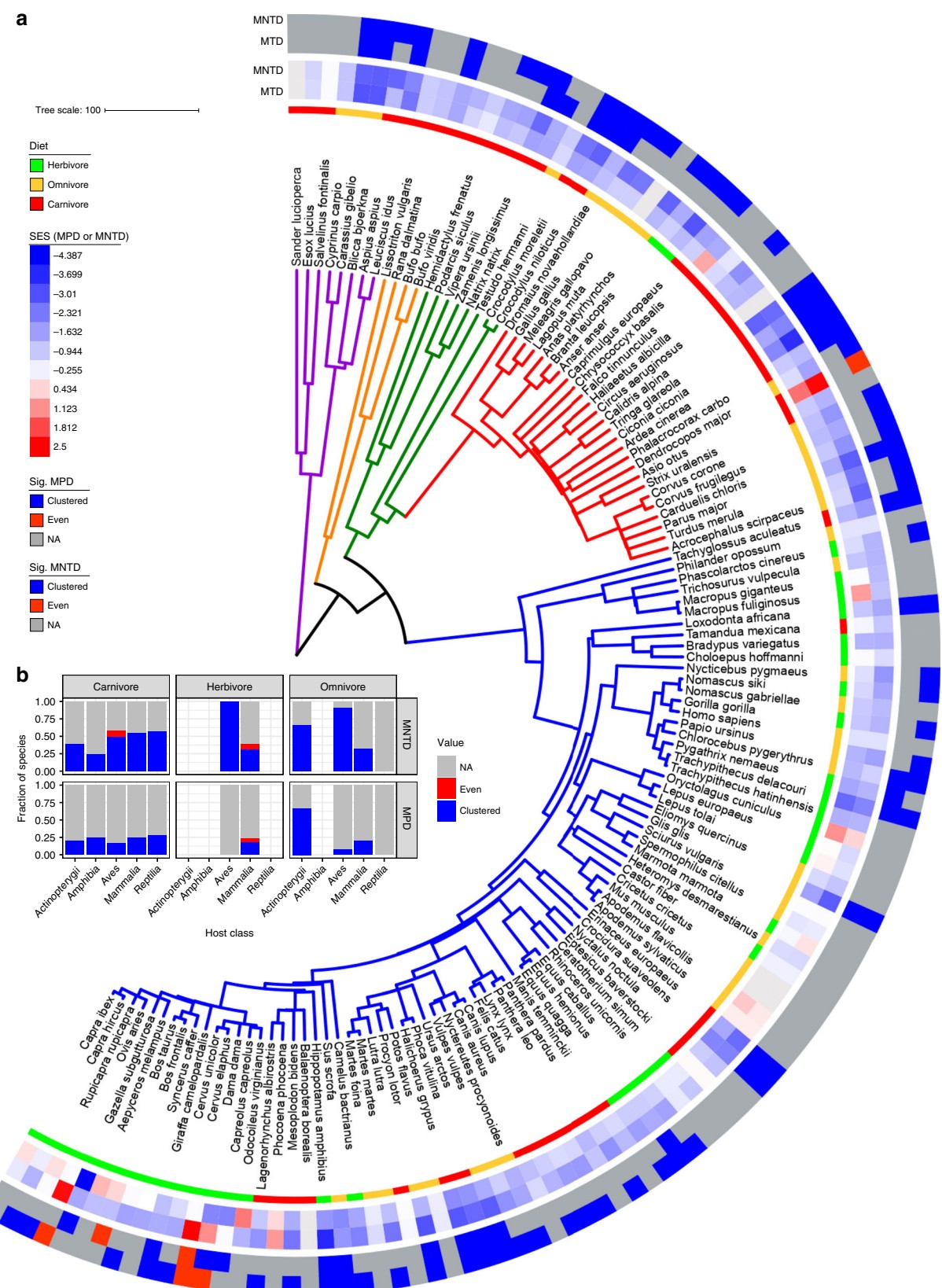

relatively abundant phylum in a number of samples (Fig. 1), but the clade has no cultured representatives and is thus poorly characterized[47]. So, as with other calls for more studies of wild animal microbiomes[9,24], our findings also advocate for more research utilizing both culture-dependent and -independent

methods to characterize the physiology, ecology, and evolution of vertebrate gut-inhabiting microbes.

While we found both host diet and evolutionary history to significantly explain microbiome diversity, each factor explained differing aspects of that diversity. At the OTU level,

**Fig. 6** Microbial communities are generally phylogenetically clustered versus evenly distributed. **a** The phylogeny is the same as shown in Fig. 1. From inner to outer, the data mapped onto the tree is host diet, mean standardized effect sizes for mean phylogenetic distance (MPD) and mean nearest taxon distance (MNTD), and samples with significant phylogenetic clustering or evenness based on MPD or MNTD. The animal species possessing microbial communities that were phylogenetically evenly distributed were the long-eared owl (*Asio otus*), fallow deer (*Dama dama*), red deer (*Cervus elaphus*), cattle (*Bos taurus*), and sheep (*Ovis aries*). **b** The bar charts depict the fraction of host species for each host class/diet where microbial taxa are more phylogenetically clustered (clustered) or evenly distributed (even) than expected from the null model or those that did not deviate from the null model (NA). Source data are provided as a Source Data file

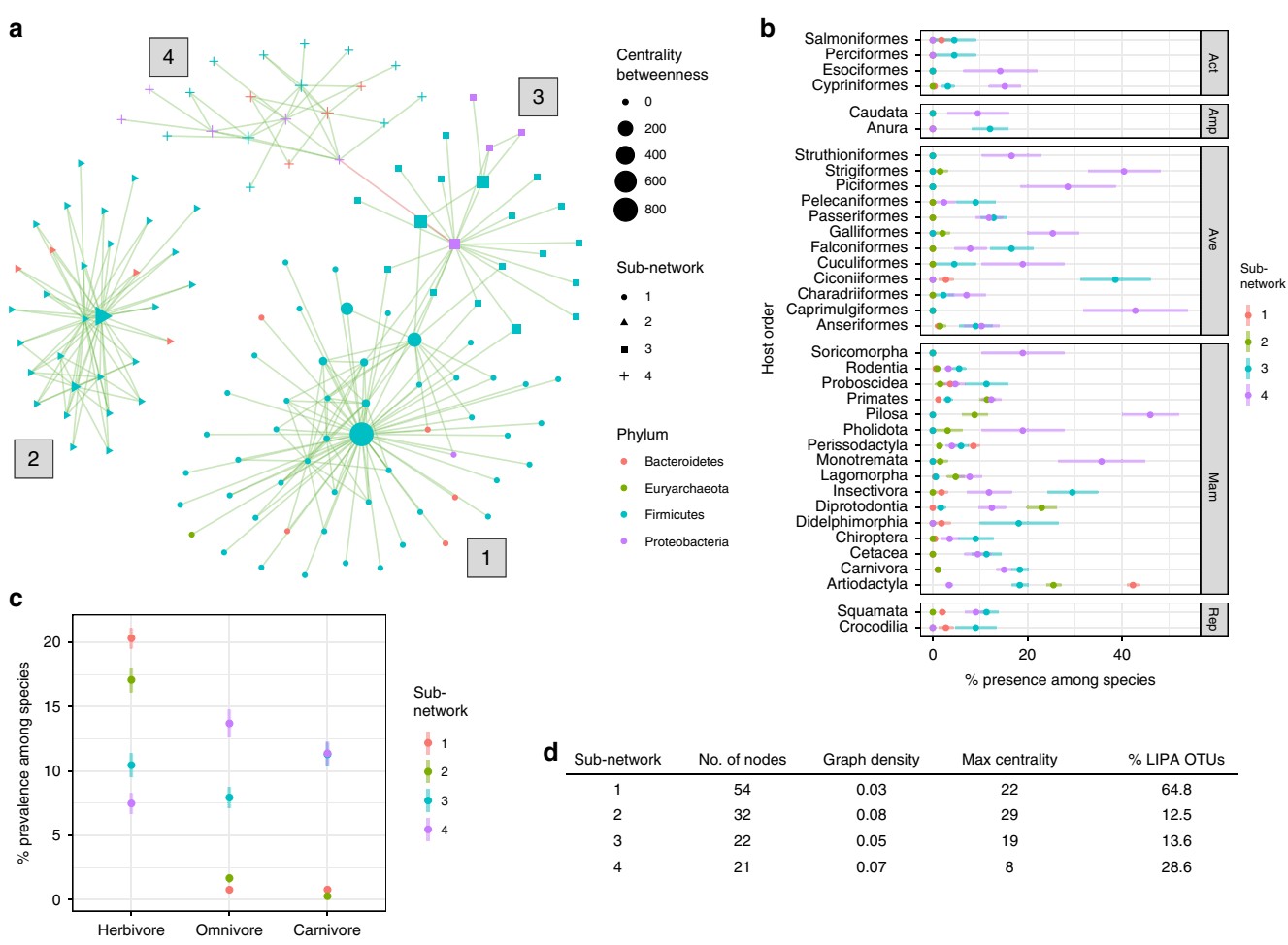

**Fig. 7** Co-occurrence analysis revealed four sub-networks of co-occurring operational taxonomic units (OTUs) that differ in taxonomic composition, network structure, and distribution across host taxonomic orders. **a** The network is a presence–absence co-occurrence network, with only significant edges shown (see "Methods"). Sub-networks are represented by differing node shapes and labels next to each sub-network. "Centrality betweenness" is a measure of how often the shortest path between two nodes transverses through the focal node. **b** The distribution of community presence among samples from each species (percentage of samples per species) shown for each host taxonomic order, with points representing the mean and line ranges representing +/− the standard error of the mean. The plots are faceted by host taxonomic class (Act = Actinopterygii, Amp = Amphibia, Ave = Aves, Mam = Mammalia, Rep = Reptilia). **c** Similar to **b** but grouped by host diet. **d** A table of sub-network statistics, with "Graph density" defined as "number of edges/total possible edges," "Max centrality" defined as the max number of shortest paths between any two nodes that cross the focal node, and "% LIPA OTUs" defined as the percentage of OTUs with significant local phylogenetic signal (Fig. 4). Source data are provided as a Source Data file

diet was a relatively strong predictor of both alpha- and beta-diversity, but the association was strongest with alpha-diversity (Figs. 2 and 3; Supplementary Fig. 17). However, the distribution of only 2 OTUs was significantly explained by diet (Fig. 3). In contrast, host phylogeny was only a significant predictor of differences in OTU composition (Fig. 2b), and 121 OTUs displayed a significant phylogenetic signal after first accounting for diet (Fig. 4).

Taken together, these results are consistent with a scenario in which diet mediates community assembly through environmental filtering predominantly at the level of functional guilds (e.g., cellulolytic consortia), while host evolutionary history mainly dictates the prevalence of specific OTUs (i.e., heritable microbial taxa). By modulating the distribution of functional guilds, host diet would increase or decrease alpha-diversity depending on the diversity of guilds selected for. If

these guilds are somewhat labile in their taxonomic composition due to functional redundancy, then the diversity of the functional guild would be dictated by diet, but taxonomic composition could vary among hosts that have the same specific diets. To illustrate, consider that a consortium degrading cellulose or other recalcitrant plant polymers in a herbivorous diet would likely require a larger assemblage of primary and secondary degraders versus a less recalcitrant meat-based diet. While microbial function can only be indirectly inferred by 16S rRNA sequencing, metagenomics studies support this concept that diet is strongly selective of microbial function, at least in the mammalian gut[25,48]. A metagenomics-based analysis on our dataset will help to resolve how diet and host phylogeny modulate microbial function versus taxonomy.

Interestingly, a few studies on mammal gut microbiomes have shown that phylogenetic signal is strongest at finer taxonomic levels, which coincides with our observations that host phylogeny mainly dictates that distribution of specific OTUs[8,21,49]. While we did observe that weighted Unifrac also became significantly associated with host phylogeny at coarser taxonomic levels (Supplementary Figs. 12, 13, 14, and 15), this may simply be the result of aggregating the abundances of multiple OTUs specific to a host species. Indeed, LIPA-OTUs associated with the same host often belonged to the same genus or family (Fig. 4). In regard to host diet, the association with microbiome beta- and alpha-diversity diminished at coarser taxonomic levels (e.g., class and phylum), but only for metrics incorporating abundance information (i.e., weighted Unifrac and Shannon Index). Indeed, many species have the same dominant phylum- and class-level taxa regardless of diet (Fig. 1), but the less abundant taxa (e.g., Spirochaetae) still show inter-diet partitioning. If diet is selecting primarily for functional guilds, then this pattern could be explained by overlap of coarse taxonomic groups among these functional guilds, especially for the more abundant taxa (e.g., Firmicutes and Bacteroidetes present in many different guilds). Our findings appear to contrast with the recent work of Groussin and colleagues[8], who found that diet mostly influences the distribution of large, ancient microbial lineages. However, their work focused on mammals in zoos, so the gut microbiome association with such artificial diets may be quite different from natural diets.

Our findings correspond with studies of microbial heritability in humans, in which the abundances of only certain specific taxonomic groups have been consistently found to be dictated by host genetics across multiple independent studies[12]. Moreover, we observed significant phylogenetic signal for OTUs belonging to all three clades identified by Goodrich and colleagues[12] to be consistently heritable in humans: *Methanobrevibacter*, *Christensenellaceae*, and *Blautia*. No OTUs in our study showed significant phylogenetic signal for humans, and only a few OTUs were associated with any of the ten primate species in our study. This finding could help to explain why relatively large cohorts are necessary to identify heritable microbial taxa in humans[12]. Alternatively, intra-species diversity is greater in large human cohort studies compared to what we measured in this work, and this higher intra-species variance may obscure signals of coevolution. Still, our findings are congruent with the work of Moeller and colleagues[50], who revealed a phylogenetic signal between African apes and gut-inhabiting Bacteroidaceae and Bifidobacteriaceae taxa. As in their study, we also observed a phylogenetic signal for Bacteroidaceae OTUs and primates. However, we found no such signal for Bifidobacteriaceae OTUs, possibly due to the low abundance of Bifidobacteriaceae in most gut microbiomes[51], which the other study overcame by using clade-targeted primers.

Both tests of phylogenetic signal at the OTU level and tests of co-speciation support the hypothesis that host evolutionary history more strongly determines microbial diversity among mammals versus non-mammals (Figs. 4 and 5; Supplementary Fig. 19). Multiple non-exclusive mechanisms could explain these findings. First, the gut microbiomes of non-mammal species may contain more transient microbes from environmental sources. This may be especially true of the Actinopterygii, given that the surrounding environment is thought to be one of the primary mechanisms of microbiota acquisition for fish[52]. Second, when considering the evolution of digestive physiology, mammals have developed highly complex digestive systems in relation to most non-mammalian species in our study[53]. This is especially true for ruminants, which have developed complex multi-chambered forestomachs and a system of regurgitation and mastication in order to efficiently degrade complex plant polymers via enhanced microbial fermentation. We observed the strongest cophylogeny signal for ruminants, especially among cattle (Bovidae), which have arguably the most complicated digestive physiology[54]. Interestingly, Nishida and Ochman found that rates of microbiome divergence have accelerated in Cetartiodactyla[21], which may be the result of evolving the complex forestomach and other digestive traits specific to this clade. Indeed, an increased microbial biomass yield for digestion by the host and increased fiber digestion are thought to represent important selective advantages in foregut fermenters[54]. Third, vertical transmission for microbial taxa from parent to offspring may also differ between mammals and non-mammals. Mammalian microbiome acquisition occurs during the birthing process and is further developed through nursing, maternal contact, and social group affiliation[55]. Much less is known about how non-mammals acquire their gut microbiomes, but at least for some species, coprophagy, eating soil in the nest, and eating regurgitated food are important modes of vertical transmission[6]. Still, mixed-mode transmission (vertical transmission and transmission from unrelated hosts or the environment) is considered to be more prevalent among non-mammals[56]. In accordance with this hypothesis, we observed a significantly higher signal of environmental biome specificity among microbial genera more associated with non-mammals versus mammals (Supplementary Fig. 20).

Our eco-phylogenetic and co-occurrence tests further resolved differences in microbial community assembly among host species. The majority of microbial communities showed significant phylogenetic clustering (Fig. 6), which supports our hypothesis that diet and host phylogeny impose environmental filtering on specific functional guilds and/or certain taxa. Interestingly, members of Artiodactyla showed little signal of phylogenetic clustering, and in some cases, we observed significant phylogenetic evenness (Fig. 6). This is consistent with the hypothesis that the effects of environmental filtering are limited among Artiodactyla. Similar observations were recently reported by Gaulke and colleagues, who found less signal of phylogenetic clustering among Artiodactyla relative to other mammalian clades[42]. The high water content of ruminant feces may help to explain this lack of phylogenetic clustering[57], given that high water content in soil has been shown to reduce phylogenetic clustering relative to dry soils[58,59]. Another non-exclusive explanatory factor may be that the refractory composition of the ruminant diet requires functional guilds composed of distantly related taxa, resulting in phylogenetic evenness. In support of this hypothesis, sub-network 1 in our co-occurrence analysis showed high specificity to Artiodactyla relative to the other sub-networks (Fig. 7), and it is the only one to contain OTUs from all four phyla present among the sub-networks (Supplementary Fig. 22).

The "hub" OTUs present in three of the four sub-networks suggests that keystone species (OTUs) contribute to community

assembly (Fig. 7). Interestingly, the maximum betweenness score in each sub-network directly corresponded with the prevalence of the sub-networks in herbivores, while the sub-network with the lowest centrality scores (sub-network 4) was the most prevalent among omnivores and carnivores (Fig. 7). Therefore, it appears that the herbivorous diet selects for co-occurring consortia containing key-stone species. These keystone species may form the foundation in which functional guilds are based. The other members of each sub-network would thus represent the taxonomically stable portion of the functional guild, while functionally redundant taxa in the guild would not show a stable co-occurrence pattern. In support of this concept, the hub OTUs of sub-networks 1 and 2 both belong to the Ruminococcaceae (Supplementary Fig. 22), and this clade contains members that can play a major role in plant cell wall breakdown into substrates utilized by other members of the consortium[60]. Indeed, Ruminococcaceae taxa have previously been identified as keystone species in human and ruminant gut communities[60]. The gain or loss of these putative keystone species in hosts may cause relatively large, diet-dependent health and fitness effects on the host.

In conclusion, our findings help to resolve the major modulators of intestinal microbiome diversity in animals, which have not been well studied in wild animals, especially non-mammalian species. Our findings indicate that diet primarily selects for functional guilds, while host evolutionary history mainly determines the prevalence of specific microbial OTUs. The modulating effect of host evolutionary history was most pronounced in mammals, especially for Artiodactyla. In general, our findings suggest that microbial community assembly in the Artiodactyla clade differs substantially from other mammalian clades, which may be the result of the complex digestive physiology that has evolved in ruminants. The putative keystone species identified in our co-occurrence analysis may be of special interest for future work determining how dietary changes can modulate the animal gut microbiome, such as in the context of captivity or climate change.

## Methods

**Sample collection.** Sampling was conducted between February 2009 and March 2014. Only fresh samples with confirmed origin from a known host species were collected, most of them by wildlife biologists conducting long-term research on the respective species in its habitat. This also ensured that sampling guidelines and restrictions were adhered to, where these were applicable. Human DNA samples were taken from a previous study[61]. Samples originated predominantly from Central Europe (Austria and neighboring countries). However, in order to cover as much vertebrate diversity as possible, many samples were also taken from other countries around the world (19 countries on 6 continents; see Supplementary Fig. 1). Detailed metadata on the sampled animal species such as habitat, sampling location, and conditions were collected alongside the fecal samples. Additional metadata was compiled from various databases such as the NCBI Taxonomy browser or the PanTHERIA database[62] (see Supplementary Data 1).

All fecal samples were collected in sterile sampling vials, transported to a laboratory and frozen within 8 h. Samples were stored at −20 °C and shipped on dry ice to TU Wien in Vienna, Austria within weeks after collection. In Vienna, DNA extraction was performed within 2 months after receiving the samples using the PowerSoil DNA Isolation Kit (MoBio Laboratories, Carlsbad, USA) in combination with bead-beating (FastPrep-24, MP Biomedicals, Santa Ana, USA). DNA concentration in extracts was measured using a NanoDrop ND 1000 UV spectrophotometer and the Quant-iT PicoGreen dsDNA Assay Kit (Thermo Fisher Scientific Inc., Vienna, Austria). DNA extracts were stored at −80 °C until further analysis.

**16S rRNA gene sequencing.** PCR amplicons for the V4 region of the 16S rRNA gene were generated with primers 515F–806R[63] and were sequenced with the Illumina MiSeq 2 × 250 v2 Kit at the Cornell University Institute for Biotechnology. DADA2[27] was used to call 100% sequence identity OTUs (i.e., sequence variants). Taxonomy was assigned to OTUs with the QIIME2 q2-feature-classifier[64] using the SILVA database (v119)[65]. The phyloseq[66] R package was used to rarefy total OTU counts to 5000 per sample due to the multiple orders of magnitude difference in raw counts among samples. A phylogeny was inferred for all OTU sequences with fas-ttree[67] based on a multiple sequence alignment generated by mafft[68]. All samples lacking metadata used in the study were filtered from the dataset. In cases where an individual host was sampled multiple times, we randomly selected one sample.

**Host phylogeny.** Only 19% of animals in our dataset have existing genome assemblies of any quality in which to infer a genome-based phylogeny from. Instead, we used a dated host phylogeny for all species from http://timetree.org[69]. To create a phylogeny for all samples (Supplementary Fig. 2), sample-level tips were grafted onto the species-level tips with a negligible branch length.

**Intra-species sensitivity analysis.** The dataset contained a variable number of samples per host species, and species were asymmetrically represented among clades (Fig. 1). Moreover, the host phylogeny did not include within-species relatedness information, which would cause zero-inflation in our analyses of coevolution. Therefore, we used a sensitivity analysis approach (inspired by the sensiphy[70] R package) that assessed the sensitivity of all analyses in this study (unless noted otherwise) to intra-species heterogeneity in microbiome diversity and host metadata. This method consisted of generating 100 subsamples of the dataset, each with just one randomly selected sample per host species. For each hypothesis tested in the study, the test was applied to each dataset subset, and the overall hypothesis test was considered significant if ≥95% of the subsets were each considered significant after correcting for multiple hypothesis testing with the Benjamini–Hochberg procedure.

**Data analysis.** General manipulation and basic analyses of the dataset were performed in R[71] with the phyloseq, dplyr, tidyr, and ggplot2 R packages[66]. High-throughput compute cluster job submission was performed with the batchtools[72] R package. Phylogenies were manipulated with the ape[73] and caper[74] R packages and visualized with iTOL[75]. Networks were manipulated and visualized with the tidygraph[76] and ggraph[77] R packages, respectively. The world map in Supplementary Fig. 1 was created with the maps[78] R package.

Similarity of OTUs to cultured representatives in the SILVA All Species Living Tree database[65] was conducted by BLASTn[79] of OTU representative sequences versus the 16S sequence database. We filtered out all BLASTn hits with an alignment length of <95% of the query sequence length. Similarity of OTUs to any representatives in SILVA was conducted in the same manner, but the BLAST database was SILVA release 132, de-replicated at 99% sequence identity.

MRMs was performed with the Ecodist[80] R package, with rank-based correlations. Effect size and significance is derived from comparing the true data to randomly permutations ($n = 1000$ for all analyses). We converted all regression variables to distance matrices through various means. The host phylogeny was represented by the patristic distance (branch lengths). We calculated the Gower distance for the detailed diet data, detailed habitat data, and sample type data (wild/captive animal + gut/feces sample origin; see Fig. 1). Geographic distance was represented as Great Circle distance based on latitude and longitude. Alpha-diversity was converted to a distance matrix by taking the Euclidean distance among all pairwise sample comparisons.

PACo[37] and Parafit[73] were performed on the host phylogeny and microbial 16S rRNA phylogeny, along with a matrix of OTU presence/absence among hosts. The Cailliez correction[81] for negative eigenvalues was applied for both PACo and Parafit. For PACo, we used the quasiswap null model, which does not assume that the symbiont is tracking the evolution of the host or vice versa (a conservative approach). For each method, 1000 permutations were used. Phylogenetic signal of OTUs was tested with the phylosignal[82] R package. Binomial regression on OTU presence/absence was used to regress out the effects of diet, and the residuals were used for tests of phylogenetic signal.

The LIPA (local Moran's I) was calculated with 9999 permutations. PGLS models were conducted with caper[74] R package. A Brownian motion model of evolution was used. For beta-diversity, the first five PCoA eigenvectors were used. Co-occurrence analyses were conducted with the cooccur[83] R package. The walktrap algorithm[84] was used for defining sub-networks in the co-occurrence network. For OTU-specific tests (LIPA, PGLS, and co-occurrence), only OTUs present in >5% of samples were included.

We downloaded the deblur_90bp.subset_2k.rare_5000 16S rRNA dataset from the EMP[38] ftp site. Samples associated with human, animal, or non-anthropogenic environmental biomes were selected ($n = 1217$) and grouped into general biome categories (e.g., "soil" or "sediment") based on EMP metadata (e.g., "env_feature"; Supplementary Data 3). Taxa were mapped between our dataset and the EMP dataset based on genus-level taxonomic classifications. For our dataset, we quantified mammal versus non-mammal specificity of genera with the IndVal analysis[39] via the labdsv R package. IndVal was also used to measure biome specificity for the EMP dataset. All genera with a Benjamini–Hochberg adjusted $p$ value of <0.05 were considered significant.

**Reporting summary.** Further information on research design is available in the Nature Research Reporting Summary linked to this article.

## Data availability

The raw sequence data are available from the European Nucleotide Archive under the study accession number PRJEB29403. All sample metadata used in this study is provided in Supplementary Data 1. All results in the manuscript can be reproduced using the metadata provided in Supplementary Data 1, the raw sequence data (PRJEB29403), and the code and notes provided on GitHub. A Source Data file is available.

## Code availability

Jupyter notebooks describing the entire data analysis process are available on GitHub at https://github.com/leylabmpi/animal_gut_16S-uni.

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

## Acknowledgements

This study was supported by the Austrian Science Fund (FWF) as part of the "Vienna Doctoral Program on Water Resource Systems" (DKplus W1219-N22) and the research project P23900 granted to A.H.F. and P22032 granted to G.H.R. Further support came from the Science Call 2015 "Resource und Lebensgrundlage Wasser" Project SC15-016 funded by the Niederösterreichische Forschungs-und Bildungsgesellschaft (NFB). This work was supported by a David and Lucile Packard Foundation Fellowship (to R.E.L) and the Max Planck Society. We would like to thank the following collaborators for their huge efforts in sample and data collection: Mario Baldi, School of Veterinary Medicine, Universidad Nacional de Costa Rica; Wolfgang Vogl and Frank Radon, Konrad Lorenz Institute of Ethology and Biological Station Illmitz; Endre Sós and Viktor Molnár, Budapest Zoo; Ulrike Streicher, Conservation and Wildlife Management Consultant, Vietnam; Katharina Mahr, Konrad Lorenz Institute of Ethology, University of Veterinary Medicine Vienna and Flinders University Adelaide, South Australia; Peggy Rismiller, Pelican Lagoon Research Centre, Australia; Rob Deaville, Institute of Zoology, Zoological Society of London; Alex Lécu, Muséum National d'Histoire Naturelle and Paris Zoo; Danny Govender and Emily Lane, South African National Parks, Sanparks; Fritz Reimoser, Research Institute of Wildlife Ecology, University of Veterinary Medicine Vienna; Anna Kübber-Heiss and Team, Pathology, Research Institute of Wildlife Ecology, University of Veterinary Medicine Vienna; Nikolaus Eisank, Nationalpark Hohe Tauern, Kärnten; Attila Hettyey and Yoshan Moodley, Konrad Lorenz Institute of Ethology, University of Veterinary Medicine Vienna; Mansour El-Matbouli and Oskar Schachner, Clinical Unit of Fish Medicine, University of Veterinary Medicine; Barbara Richter, Institute of Pathology and Forensic Veterinary Medicine, University of Veterinary Medicine Vienna; Hanna Vielgrader and Zoovet Team, Schönbrunn Zoo; Reinhard Pichler, Herberstein Zoo. We explicitly thank the Freek Venter of South African National Parks and the National Zoological Gardens of South Africa for granting access to their Parks for sample collection. We also thank Carolin Kolmeder and Jillian Waters for their helpful discussions on this project.

## Author contributions

G.H.R., R.E.L., and A.H.F. created the study concept. G.H.R., N.S., C.W., and G.S. performed the sample collection and metadata compilation. G.H.R. and N.S. performed the laboratory work. N.D.Y. and W.W. performed the data analysis. N.D.Y., G.H.R., R.E.L., and A.H.F. wrote the manuscript.

## Additional information

**Competing interests:** The authors declare no competing interests.

