## [Peer Review File · Nature Communications]

Reviewers' Comments:

Reviewer #1:

Remarks to the Author:

This paper presents a new dataset of (mainly) wild animal gut microbiomes. Data consist of 16S amplicons, were sampled across a large number of host species, which cover 5 classes of vertebrates. This new dataset greatly increases the representation of Vertebrate species, with an interesting inclusion of many non-mammalian species – which are underrepresented and under-studied in the literature. Sampling and generating all these data represented a huge amount of work, and I congratulate the authors for this effort. I also very much appreciate the fact that the authors uploaded their scripts on github, which greatly contributes to transparency and reproducibility of their research.

Overall, I found that most of the analyses are sound, but not very novel. In particular, the newly sampled non-mammalian Vertebrates are not very well exploited. Maybe the reason is because its main focus, namely the influence of host diet and phylogeny on microbiome compositions, has already been investigated several times in the past, and that the present analyses do not add much to what was reported before. However, I strongly believe that there is great potential for this paper, and I hope that the authors will find my comments below helpful.

Major comments

The fact that non-mammalian species exhibit weaker correlation with phylogeny is very interesting and not very well understood in the literature. The authors suggest that this may be due to the higher presence of transient bacteria from environmental sources in non-mammalian species as compared to mammalian ones (L576). This is a very appealing hypothesis, but the manuscript left me a bit hungry for more, such as a proper test of this hypothesis. It's possible to map the 16S data of non-mammalian species to 16S data obtained from environmental sources (water, soil, oceans, etc..) that are present in public databases, and compare to what extent similarity is higher for non-mammalian species than for mammalian species. This would constitute a very novel result regarding what is known in the literature. As I know that doing new analyses is time consuming, I left it open for the authors to do it or not, depending on whether they are convinced or not that this would increase the interest of their paper.

The fact that the authors did not find association between microbiome compositions and geographic distance is interesting. The authors recognize that this is in contrast with some of the previous reports showing that sympatric mammals share more microbes than expected by chance or phylogeny (See Moeller et al., PNAS 2017 & Moeller et al. Genome Res 2013). I think the authors should investigate or discuss more extensively the origin of these inconsistencies. Especially because the authors claim that their experimental/sampling approach is not designed to tackle this question (L277), but I don't see any reason why (considering that they have animals living in various areas around the world, some being very close geographically). Have the authors looked whether the signal is stronger when only looking at mammals? Only looking at animals that are known to interact across predator-prey relationships?

Measures of correlation between microbiome compositions and host phylogeny or diet, as well as the detection of individual OTUs that have distributions across hosts that correlate with these two factors have been done at a single, very recent, bacterial taxonomic level, namely dada2 OTUs, which are at the tips of the bacterial tree. However, previous reports showed that the effect of phylogeny and diet are different across bacterial phylogenetic resolutions, correlation with phylogeny deriving from recent bacterial evolution, while diet is associated with the presence or absence of broader, more ancient, bacterial clades (Sanders et al, Mol Ecol 2014 & Groussin et al Nat Comm 2017). These previous

reports suggested that diet environmentally filters for high-level clades whose lineages all share similar functions that are important to metabolize diet material, consistent with the conservation of functional classes of enzymes involved in, say, cellulose degradation, across multiple lineages of the same class or orders of bacteria. This interpretation is also suggested by the authors, L315-328, and discussed more in length at L544-556, but is not supported by analyses that make use of their interesting new data. This would be an interesting analysis to do to see if interpretations hold true with more comprehensive data. The authors could then discuss the model presented at L544-556 in the context of this previous literature.

The authors test for co-phylogeny between bacteria and hosts, which is a fascinating question. The authors should discuss their results in the context of what was previously found in two recent papers that tested for the same thing (Moeller et al, Science 2016, Groussin et al Nat Comm 2017), and which found bacterial lineages that harbor significant signals for co-phylogeny/co-speciation.

L253: It is very unusual in the field of community ecology to use MRM with alpha diversities, which are encoded in vector-type data formats. The authors had to convert the raw alpha diversities into distance data in a matrix format to use MRM, while correlations between raw alpha diversity and host phylogeny or diet could have been performed with a PERMANOVA approach. Note also that converting the alpha diversities into matrix-type format depends on the choice of the distance metric, which can make you lose information if it is inadequate. I don't have a very strong opinion against it, but the authors should motivate a little more this unusual approach.

Minor comments

Unless I'm wrong, the methods do not explain what dietary metadata were used in this study. Is it metadata recorded from observations in the wild or did the authors use already compiled dietary databases for vertebrates? Supplementary files show that it is coarse-grained information. This is fine, but it would be interesting to discuss the granularity of the dietary information and its potential impact on results when comparing diet to phylogeny.

I appreciate the effort to measure the effect of inter-individual variation in microbiome compositions on their results, and the sensitivity analysis that the authors have used is sound. After looking at Supp Fig 2, one can see quite a lot of inter-individual variation in relative abundances of the main phyla for hosts that have individual replicates. In an NMDS plot, do you see individuals clustering by species when weighted-beta diversities are measured? I'm basically wondering whether the fact that phylogeny is not significantly correlated to weighted beta diversity is not only explained by variations in abundance-weighted compositions among closely related-species, but also by variations across individuals within each species.

L92-94 and Results: it's also possible that some OTUs respond to both factors, i.e. are associated with diet and have a distribution restricted to some very specific host clades. Have you identified these OTUs and quantified their amount? It would be interesting to have these estimates in order to evaluate to what extent phyllosymbiosis can be generated by similar diet preferences among closely-related hosts.

I find the first Result section (Concept of sampling) a little weak. It could be moved to the Methods. I leave it up to the authors, but I think that it could help raising the interest of the reader when starting the Results.

I also find that the second section (Low prevalence and limited representation of cultured isolates) is not very well motivated, and agglomerates different analyses and high-level discussions on the data

that are not centered on a clear question. I would advise the authors to re-organize these results in order to increase the impact of their main analyses and results.

L157: By default, the MRM function in the ecodist package computes a null model and compares the observed R2 to this null. However, I think that the authors could clarify the explanation of their statistical protocol, as I originally thought that the p-values were computed from comparing the observed R2 to the R2 distribution obtained from the Intra-species sensitivity analysis. (I had to check the MRM documentation to make sure that MRM does indeed a comparison to a null). So it may be useful to explain that the observed R2s are first compared to a null in ecodist, explain what this null is (there are dozens of possible null distributions that one can investigate depending on how you shuffle/rearrange the matrices), and then explain that the sensitivity analysis comes at last, to measure the impact of within-species variation.

Other comments

L60: magnitude these two -> magnitude of these two

L64: Reference 19 is about skin microbiome, while the whole paragraph discusses the gut microbiome. Please correct using a more appropriate reference.

L101: do you also confirm that each sample comes from a different individual? It is sometimes difficult in zoos or in the wild to accurately identify the source of a stool sample, so it would be helpful for the reader to be sure that all samples within each host species truly represent different individuals, and not replicates of the same individuals.

L110: "transported to a laboratory" - on ice?

L167: provide a reference for the Cailliez correction.

L175: conducted with -> were conducted with the

L178: The walktrap algorithm for -> The walktrap algorithm was used for

Figure 1: Canis lupus has diet info colored in Brown, while legend colors are black, orange or yellow.

L331: "In contrast to the PGLS analysis" - What do you mean? In contrast to what was found with diet?

L383: with model -> with a model

Reviewer #2:

Remarks to the Author:

The study by Youngblut et al is a novel and well-written assessment of a key question in microbiome research. Specifically, the goal of this study was to test the contributions of diet and evolution in shaping host gut microbial communities across a wide range of host species. To do so, the study was carefully designed to avoid technical confounds; primarily wild animals were used to avoid captivity effects, all samples were sequenced together to avoid batch effects, and 213 samples were collected from 128 host species across 5 classes (Mammalia, Aves, Reptilia, Amphibia, and Actinopterygii) to create a taxonomically broad dataset. Further, the statistics were carefully run, incorporating sensitivity analyses to control for variation in the number of samples per host species. The authors found that diet predicted several metrics of overall microbiome composition but the presence of only two OTUs, while host phylogeny had little effect on overall microbiome composition but predicted the distribution of many OTUs. Further, they found that OTU co-occurrence networks varied across host diets and host taxonomic groups. Taken together, these results provide a valuable contribution to the microbiome literature - specifically, the finding that diet and host species explain different aspects of the microbiome is novel, However, there are several issues that should be addressed:

Major issues

1. The introduction raises 2 key issues this study seeks to control for, one of which is the difference between captive and wild animals' microbiota. But, this study still uses 20% captive animals. Although, contrary to other work, they found no effect of captivity, it might still be useful to analyze the data excluding the captive animals, and validate that the signal does not disappear.
2. The second key issue the study aimed to control for is that "Host phylogenies are inferred from differing molecular data" (L80). However, the host phylogenetic data is obtained from timetree.org, a database that calculates branch lengths and divergence times by amalgamating various methods from different studies. Because a major goal of the paper is to test the importance of host phylogeny, it would be helpful if the authors added an argument for why calculating phylogenetic distances for different sections of their host tree based on different methods is a strong approach. It may also be useful to analyze the data using phylogenetic distance that is based on genetic data, such as from the UCSC multiple genome alignments, using the subset of species for which genetic data is available.
3. There is little information on sample size in the main text. This is a key element, and including the sample sizes in each diet type and clade, especially in the analyses that compare Artiodactyls to other taxonomic groups, would strengthen the arguments. For example, supplementary figures S1a and S1b would be a useful main text figure.
4. Some of the conclusions extrapolate further than can be supported given the study design. For example, the authors state their results suggest that diet differences select for functional guilds of microbes (e.g., L316, L544-556, L634) but a functional analysis is not conducted. It is also suggested that the presence of microbial taxa indicate that they specially adapted to particular hosts (L362-364, L516-524) and that there are no effects of evolutionary history on OTU presence in non-mammalian hosts (L337). It might be useful to tone down these conclusions or provide additional arguments.

Minor comments

Introduction

L63-65 Give examples of clades

Methods

OTU prevalence analyses were run on data rarefied to 5,000 reads per sample. What was the average number of OTUs per sample in the full dataset versus the rarefied dataset? Would running models on a transformed, rather than rarefied, dataset yield different results?

Results

L201- Add the maximum number of samples per species- how many had >1 sample?

Fig. 1 does a nice job of visualizing a lot of information, but it would be helpful to add more detailed diet definitions- what do 0, 0.5, and 1 mean? Also, it looks like *Nyctalus noctula* and *Crocidura suaveolens* (bottom left) samples are completely Proteobacteria (sequencing error or contamination?).

L209- What OTU subsample depth is used to calculate this average? <5,000 reads?

L215- What percentage of OTUs were in >5% of samples?

Discussion

L470- Are the subnetwork shifts among diets statistically significant?

L548- "would expand or contract alpha-diversity". It would be clearer to say increase or decrease.

L587- "increased microbial yield and fiber digestion". Sentence is unclear. Do you mean greater microbial richness and ability to digest fiber?

L602- "This is consistent with a hypothesis that the effects of environmental filtering are limited among Artiodactyla compared to processes selecting for unrelated taxa." Sentence reads as though processes are deliberately selecting for unrelated taxa.

Reviewers' comments

Note: point-by-point responses are in bold

Reviewer #1 (Remarks to the Author):

This paper presents a new dataset of (mainly) wild animal gut microbiomes. Data consist of 16S amplicons, were sampled across a large number of host species, which cover 5 classes of vertebrates. This new dataset greatly increases the representation of Vertebrate species, with an interesting inclusion of many non-mammalian species – which are underrepresented and under-studied in the literature. Sampling and generating all these data represented a huge amount of work, and I congratulate the authors for this effort. I also very much appreciate the fact that the authors uploaded their scripts on github, which greatly contributes to transparency and reproducibility of their research.

Overall, I found that most of the analyses are sound, but not very novel. In particular, the newly sampled non-mammalian Vertebrates are not very well exploited. Maybe the reason is because its main focus, namely the influence of host diet and phylogeny on microbiome compositions, has already been investigated several times in the past, and that the present analyses do not add much to what was reported before. However, I strongly believe that there is great potential for this paper, and I hope that the authors will find my comments below helpful.

Major comments

The fact that non-mammalian species exhibit weaker correlation with phylogeny is very interesting and not very well understood in the literature. The authors suggest that this may be due to the higher presence of transient bacteria from environmental sources in non-mammalian species as compared to mammalian ones (L576). This is a very appealing hypothesis, but the manuscript left me a bit hungry for more, such as a proper test of this hypothesis. It's possible to map the 16S data of non-mammalian species to 16S data obtained from environmental sources (water, soil, oceans, etc..) that are present in public databases, and compare to what extent similarity is higher for non-mammalian species than for mammalian species. This would constitute a very novel result regarding what is know in the literature. As I know that doing new analyses is time consuming, I left it open for the authors to do it or not, depending on whether they are convinced or not that this would increase the interest of their paper.

We agree with the reviewer that this analysis could prove to be very insightful. A comprehensive analysis would likely extend beyond the scope of this work and thus require a separate publication. We did perform an initial analysis, which can later be

expanded upon in future work. More specifically, we identified which taxa were specific to either mammals or non-mammal hosts via the indicator value analysis (IndVal). We mapped these taxa to a large 16S dataset of samples from the Earth Microbiome Project (mapping done at the Genus level; see the newly added Supplementary Table 3 for a list of samples used). We then determined which of the mammal-specific or non-mammal-specific genera were associated with environmental and/or human/animal biomes in the EMP dataset (also based on IndVal). As we hypothesized, the non-mammal-specific genera showed a stronger signal of environmental biome association versus mammal-specific genera, and this signal was significant (Wilcox: $P < 0.006$). Our new findings are shown in Supplementary Fig. 20, and we have updated the Methods, Results, and Discussion text to include this new analysis.

The fact that the authors did not find association between microbiome compositions and geographic distance is interesting. The authors recognize that this is in contrast with some of the previous reports showing that sympatric mammals share more microbes than expected by chance or phylogeny (See Moeller et al., PNAS 2017 & Moeller et al. Genome Res 2013). I think the authors should investigate or discuss more extensively the origin of these inconsistencies. Especially because the authors claim that their experimental/sampling approach is not designed to tackle this question (L277), but I don't see any reason why (considering that they have animals living in various areas around the world, some being very close geographically). Have the authors looked whether the signal is stronger when only looking at mammals? Only looking at animals that are known to interact across predator-prey relationships?

We do recognize that our lack of association between gut microbiome diversity and geographic distance contrasts with some recent studies, and we have included this information in the Discussion text.

The experimental design conflates geographic location with animal relatedness, diet, and the other cofactors. An experimental design that would have more power for detecting microbiome ~ geography associations would include more samples that better decouple these covariates (e.g., same diet, different location or similar/same species, different location). We have edited the text in the Results and Discussion to try and clarify this point.

We conducted the MRM analysis on just mammal samples and found essentially the same results as when assessing the entire dataset (Supplementary Fig. 11), suggesting that the gut microbiomes of both mammals and non-mammals do not have a substantial association with geography. In regards to investigating predator-prey relationships, we would only be able to use theoretical predator prey pairs, and not necessarily pairs overlapping in habitat ranges. Again, the experimental design was not aimed at discerning these relationships, thus leading to very few geographically overlapping (theoretical) predator/prey pairs in our dataset . Therefore, we are doubtful that such an analysis would lead to true insights, and thus we would prefer to exclude it from the scope of this work.

Measures of correlation between microbiome compositions and host phylogeny or diet, as well as the detection of individual OTUs that have distributions across hosts that correlate with these two factors have been done at a single, very recent, bacterial taxonomic level, namely dada2 OTUs, which are at the tips of the bacterial tree. However, previous reports showed that the effect of phylogeny and diet are different across bacterial phylogenetic resolutions, correlation with phylogeny deriving from recent bacterial evolution, while diet is associated with the presence or absence of broader, more ancient, bacterial clades (Sanders et al, Mol Ecol 2014 & Groussin et al Nat Comm 2017). These previous reports suggested that diet environmentally filters for high-level clades whose lineages all share similar functions that are important to metabolize diet material, consistent with the conservation of functional classes of enzymes involved in, say, cellulose degradation, across multiple lineages of the same class or orders of bacteria. This interpretation is also suggested by the authors, L315-328, and discussed more in length at L544-556, but is not supported by analyses that make use of their interesting new data. This would be an interesting analysis to do to see if interpretations hold true with more comprehensive data. The authors could then discuss the model presented at L544-556 in the context of this previous literature.

In order to investigate how host diet and phylogeny associate with broader taxonomic lineages, we performed additional MRM analyses with OTUs aggregated at various taxonomic levels ranging from genus to phylum (Supplementary Fig. 12, 13, 14, & 15). We have added our new findings to the Results. Also, we compare and contrast our results with the work of Nishida and Ochman Mol. Eco. 2018, and Groussin and colleagues Nat. Comm. 2017 in the Discussion.

The authors test for co-phylogeny between bacteria and hosts, which is a fascinating question. The authors should discuss their results in the context of what was previously found in two recent papers that tested for the same thing (Moeller et al, Science 2016, Groussin et al Nat Comm 2017), and which found bacterial lineages that harbor significant signals for co-phylogeny/co-speciation.

We agree that a more elaborate comparison of our findings to those of Moeller et al, 2016 and Groussin et al 2017 would be beneficial. Therefore, we expanded our existing comparison of our findings to these other two studies in the Discussion.

L253: It is very unusual in the field of community ecology to use MRM with alpha diversities, which are encoded in vector-type data formats. The authors had to convert the raw alpha diversities into distance data in a matrix format to use MRM, while correlations between raw alpha diversity and host phylogeny or diet could have been performed with a PERMANOVA approach. Note also that converting the alpha diversities into matrix-type format depends on the choice of the distance metric, which can make you lose information if it is inadequate. I don't

have a very strong opinion against it, but the authors should motivate a little more this unusual approach.

We feel that using MRM on alpha-diversity is appropriate in this context for multiple reasons. First, many of the covariates are distances (e.g., host phylogeny and geographic distance), which MRM directly utilizes. We could utilize dimension reduction techniques (e.g., PCoA) to convert these distances to vectors, as would be needed for standard multiple regression, but we would have to select only certain reduced dimensions (e.g., PC1 and PC2), and this excludes some of the (potentially important) variance from the analysis. Alternatively, we could cluster the distance-based covariates to create discrete factors, but that will again remove some of the variance from the analysis and could exacerbate treatment group imbalances (e.g., mammalia taxonomic groups are over-represented). Second, utilizing MRM on alpha-diversity allows for a direct comparison between alpha- and beta-diversity, which reduces the possibility that methodological differences caused different findings when assessing alpha- versus beta-diversity (e.g., we found significant associations between host phylogeny and beta-diversity but not alpha-diversity).

Minor comments

Unless I'm wrong, the methods do not explain what dietary metadata were used in this study. Is it metadata recorded from observations in the wild or did the authors use already compiled dietary databases for vertebrates? Supplementary files show that it is coarse-grained information. This is fine, but it would be interesting to discuss the granularity of the dietary information and its potential impact on results when comparing diet to phylogeny.

We used metadata already compiled in databases for vertebrates, mainly from PanTHERIA (Jones et al., Ecology 2009). Where the database yielded no information, we supplemented with data coming directly from the scientist working with the animals in the field. We changed the wording in the Methods section to make this clearer.

I appreciate the effort to measure the effect of inter-individual variation in microbiome compositions on their results, and the sensitivity analysis that the authors have used is sound. After looking at Supp Fig 2, one can see quite a lot of inter-individual variation in relative abundances of the main phyla for hosts that have individual replicates. In an NMDS plot, do you see individuals clustering by species when weighted-beta diversities are measured? I'm basically wondering whether the fact that phylogeny is not significantly correlated to weighted beta diversity is not only explained by variations in abundance-weighted compositions among closely related-species, but also by variations across individuals within each species.

We agree with the reviewer that intra-species variability in microbiome diversity may have reduced our power to detect associations between host phylogeny and

weighted Unifrac. To assess this possibility, we measured the beta dispersion (distance from multivariate centroid) for both weighted and unweighted Unifrac metrics (Supplementary Fig. 6). If intra-species microbiome variability appears greater when assessed with a weighted beta-diversity metric, then beta dispersion should be greater for weighted versus unweighted Unifrac. Instead, we found less intra-species variance for weighted Unifrac relative to unweighted Unifrac (Supplementary Fig. 6). Therefore, we postulate that the differences between the two metrics in regards to an association with host phylogeny is the result of host phylogeny mainly dictating microbial community composition, but not OTU abundances. We have updated the Results to include these new findings and added the new supplementary figure (Supplementary Fig. 6).

L92-94 and Results: it's also possible that some OTUs respond to both factors, i.e. are associated with diet and have a distribution restricted to some very specific host clades. Have you identified these OTUs and quantified their amount? It would be interesting to have these estimates in order to evaluate to what extent phyllosymbiosis can be generated by similar diet preferences among closely-related hosts.

We agree with the reviewer that some OTUs could possibly respond to both diet and phylogeny. However, we did not find any OTUs that associated with both diet (Fig. 3) and host phylogeny (Fig. 4). This does not exclude the possibility of such OTUs existing, and possibly a more powered study will reveal them.

I find the first Result section (Concept of sampling) a little weak. It could be moved to the Methods. I leave it up to the authors, but I think that it could help raising the interest of the reader when starting the Results.

We are aware that the initial part of the Results does not present any exciting findings. Nevertheless, we think that the careful and very deliberate study design and sample selection is one of the main factors setting this work apart from previous studies in this field. The general exclusion of zoo animals and the effort to represent a broad range of phylogenetic groups are steps towards a more representative picture of intestinal community composition in vertebrates . We also think that the reader needs the information represented here to be able to judge the relevance of the results of the microbiota analysis. We shortened the section slightly but would prefer to keep it here at the beginning of the results section.

I also find that the second section (Low prevalence and limited representation of cultured isolates) is not very well motivated, and agglomerates different analyses and high-level discussions on the data that are not centered on a clear question. I would advice the authors to re-organize these results in order to increase the impact of their main analyses and results.

We believe that this section provides a great deal of valuable information that is necessary for proper critical evaluation of the analyses performed in subsequent sections of the manuscript. First, we find that OTUs are sparsely distributed, which justifies our use of a binary transformation for many of the subsequent analysis. Second, we find that host phylogeny constrains beta-diversity, which provides the first evidence of an association between host phylogeny and beta-diversity. Third, we validate our 16S rRNA sequencing and sequence data processing by showing that the microbiome taxonomic composition of many host species in our study matches findings from other animal gut microbiome studies. Fourth, we show that many OTUs in our dataset, especially those observed in non-mammals, lack cultured representatives. We believe this last finding is very important for directing future efforts for culturing these as-of-yet uncultured microbes. We have included this justification at the end of this subsection in the Results in order to emphasize the importance of these findings.

L157: By default, the MRM function in the ecodist package computes a null model and compares the observed R2 to this null. However, I think that the authors could clarify the explanation of their statistical protocol, as I originally thought that the p-values were computed from comparing the observed R2 to the R2 distribution obtained from the Intra-species sensitivity analysis. (I had to check the MRM documentation to make sure that MRM does indeed a comparison to a null). So it may be useful to explain that the observed R2s are first compared to a null in ecodist, explain what this null is (there are dozens of possible null distributions that one can investigate depending on how you shuffle/rearrange the matrices), and then explain that the sensitivity analysis comes at last, to measure the impact of within-species variation.

We agree with the reviewer that our analysis is complex, and somewhat novel, and therefore it requires a greater explanation of the methodological details. We have added more methodological details on this analysis in the Methods in order to clarify the details of our procedure.

Other comments

L60: magnitude these two -> magnitude of these two

We have fixed this typo in the manuscript.

L64: Reference 19 is about skin microbiome, while the whole paragraph discusses the gut microbiome. Please correct using a more appropriate reference.

We removed this citation but did not include another, since we believe that the other two citations are sufficient.

L101: do you also confirm that each sample comes from a different individual? It is sometimes difficult in zoos or in the wild to accurately identify the source of a stool sample, so it would be helpful for the reader to be sure that all samples within each host species truly represent different individuals, and not replicates of the same individuals.

We can confirm that each specific sampling site was only sampled once for this study. That is also the case for zoo enclosures. There was repeated sampling in some cases in one compound but these samples were actually taken from marked individual animals directly after defecation. So, unintentional resampling of the same individual can be excluded.

L110: “transported to a laboratory” – on ice?

We can not assert that that was the case with all the samples. In some very remote sampling areas, transport in cooling boxes might not have been feasible. Nevertheless sample collectors were required to cool the samples as soon as possible and also freeze them as soon as they reached a lab. Some delay between the excretion and the time of conservation can not be avoided in any case.

L167: provide a reference for the Cailliez correction.

We have added the appropriate citation.

L175: conducted with -> were conducted with the

We have fixed this typo in the manuscript.

L178: The walktrap algorithm for -> The walktrap algorithm was used for

We have fixed this typo in the manuscript.

Figure 1: Canis lupus has diet info colored in Brown, while legend colors are black, orange or yellow.

We have added “0.25” color (brown) to the legend in order to help clarify that the colors correspond to averages among all individuals of that species in our dataset (e.g.,

detailed diet components sometimes differed among individuals of the same species). See also the response to Reviewer 2 on this same topic.

L331: "In contrast to the PGLS analysis" - What do you mean? In contrast to what was found with diet?

Yes, the LIPA findings suggest little phylogenetic signal of alpha diversity, which contrasts the substantial and significant associations of host diet and alpha diversity (as seen with PGLS). The text in this paragraph have been updated to clarify this point.

L383: with model -> with a model

We have fixed this typo in the manuscript.

Reviewer #2 (Remarks to the Author):

The study by Youngblut et al is a novel and well-written assessment of a key question in microbiome research. Specifically, the goal of this study was to test the contributions of diet and evolution in shaping host gut microbial communities across a wide range of host species. To do so, the study was carefully designed to avoid technical confounds; primarily wild animals were used to avoid captivity effects, all samples were sequenced together to avoid batch effects, and 213 samples were collected from 128 host species across 5 classes (Mammalia, Aves, Reptilia, Amphibia, and Actinopterygii) to create a taxonomically broad dataset. Further, the statistics were carefully run, incorporating sensitivity analyses to control for variation in the number of samples per host species. The authors found that diet predicted several metrics of overall microbiome composition but the presence of only two OTUs, while host phylogeny had little effect on overall microbiome composition but predicted the distribution of many OTUs. Further, they found that OTU co-occurrence networks varied across host diets and host taxonomic groups. Taken together, these results provide a valuable contribution to the microbiome literature - specifically, the finding that diet and host species explain different aspects of the microbiome is novel. However, there are several issues that should be addressed:

Major issues

1. The introduction raises 2 key issues this study seeks to control for, one of which is the difference between captive and wild animals' microbiota. But, this study still uses 20% captive

animals. Although, contrary to other work, they found no effect of captivity, it might still be useful to analyze the data excluding the captive animals, and validate that the signal does not disappear.

The reviewer raised a good point that we could do more to assess the effects of captivity vs wild besides including the information as a covariate in the MRM analysis. We performed another MRM analysis with just wild animals (total samples = 170; total host species = 119), with the “sample type” covariate now just determining fecal vs gut samples. We found no substantial difference relative to the MRM analysis on the full dataset (Supplementary Fig. 10).

2. The second key issue the study aimed to control for is that “Host phylogenies are inferred from differing molecular data” (L80). However, the host phylogenetic data is obtained from timetree.org, a database that calculates branch lengths and divergence times by amalgamating various methods from different studies. Because a major goal of the paper is to test the importance of host phylogeny, it would be helpful if the authors added an argument for why calculating phylogenetic distances for different sections of their host tree based on different methods is a strong approach. It may also be useful to analyze the data using phylogenetic distance that is based on genetic data, such as from the UCSC multiple genome alignments, using the subset of species for which genetic data is available.

We assessed whether creating a phylogeny based on host genomes was feasible for our dataset. Even when including available genomes from both the UCSC and also Genbank, <19% of the host species in our study have available genomes of any quality. Therefore, we do not believe this approach is viable for our study.

In regards to the introduction of our manuscript, we point out challenges in this field, but we must admit that we cannot fully address these challenges in our study. For instance, besides using an ensemble tree synthesized from thousands of individual trees, we also have a dataset that is biased toward mammals. In this work, we tried to address these challenges as best as possible. For example, we utilize a host phylogeny instead of just relying on taxonomic groupings of host species, which helps to alleviate biases in representation and allows for a more nuanced analysis via employing continuous variables. We believe it is this section of the Introduction is important for highlighting the challenges in this field, given that we do then go on to show how we are imperfectly addressing those challenges in our study.

3. There is little information on sample size in the main text. This is a key element, and including the sample sizes in each diet type and clade, especially in the analyses that compare Artiodactyls to other taxonomic groups, would strengthen the arguments. For example, supplementary figures S1a and S1b would be a useful main text figure.

We have combined S1a and S1b with Fig. 1. We have also modified the Results to include more information on sample sizes.

4. Some of the conclusions extrapolate further than can be supported given the study design. For example, the authors state their results suggest that diet differences select for functional guilds of microbes (e.g., L316, L544-556, L634) but a functional analysis is not conducted. It is also suggested that the presence of microbial taxa indicate that they specially adapted to particular hosts (L362-364, L516-524) and that there are no effects of evolutionary history on OTU presence in non-mammalian hosts (L337). It might be useful to tone down these conclusions or provide additional arguments.

We agree with the reviewer that we did not directly assess function via metagenomics or other means, and therefore we cannot make direct conclusions about microbial functioning. While our findings do not fully test our hypothesis of diet selecting for functional guilds (and host evolutionary mainly dictating the distributions of certain OTUs), we argue that these findings are *consistent* with this hypothesis. We have carefully worded our conclusive statements in the manuscript in order to not overly inflate the implications of our findings. Moreover, we do state in the Results that a metagenomics-based study is needed to further test our hypothesis. Although some would argue that we could perform functional analysis with our current 16S rRNA data (e.g., by using Picrust), we do not believe that such methods are robust enough for the caliber of this study.

Minor comments

Introduction

L63-65 Give examples of clades

We have added multiple example of clades to this portion of the text.

Methods

OTU prevalence analyses were run on data rarefied to 5,000 reads per sample. What was the average number of OTUs per sample in the full dataset versus the rarefied dataset? Would running models on a transformed, rather than rarefied, dataset yield different results?

While the mean post-QC counts per sample were >40,000, many of the non-mammalian samples had much lower sampling depths that were closer to 5,000. So to prevent loss of these samples via rarefaction at a higher value, we chose 5,000 as the rarefaction depth.

We did investigate the use of multiple transformations, as these can often be a better alternative to throwing away data via rarefying. However, our dataset differs from

many microbiome studies due to the diversity of samples (hosts) included. Unlike human gut microbiome studies, which generally have substantial overlap of OTUs among individuals, we are assessing gut microbiomes from a large phylogenetic diversity of host species that can differ greatly in diets, ecologies, geographic locations, etc. The diversity of our microbiome dataset samples likely resulted in the low overlap of OTUs across samples (i.e., high OTU sparsity). The high percentage of zero abundances does not work well with most/all transformations that have been shown to at least partially alleviate compositional effects (i.e., distortions of ‘true’ abundances resulting from using relative abundances). More specifically, these transformations often do not work with zero counts due to employing ratios in the algorithm. Thus, a pseudo-count must be used (e.g., convert all zero counts to 1). Utilizing pseudo-counts can dramatically distort the dataset by “raising up” all 0 count OTUs to the same baseline abundance (e.g., all at a count of 1), even though these taxa could differ in their actual abundances by orders of magnitude. Indeed, we found that applying such transformations to our dataset resulted in unrealistic beta-diversity patterns. For these reasons, we believe that our use of rarefaction was appropriate for this particular microbiome dataset.

Results

L201- Add the maximum number of samples per species- how many had >1 sample?

Fifty species had >1 sample. We have added this information to this section. This information can also be visually assessed in Supplementary Fig. 2.

Fig. 1 does a nice job of visualizing a lot of information, but it would be helpful to add more detailed diet definitions- what do 0, 0.5, and 1 mean? Also, it looks like *Nyctalus noctula* and *Crocidura suaveolens* (bottom left) samples are completely Proteobacteria (sequencing error or contamination?).

The “Diet (detailed)” information varies among some individuals, and the values shown are averages of the binary yes/no values for each individual. We have added this information to the figure legend. See also the response to Reviewer 1 on this same topic.

L209- What OTU subsample depth is used to calculate this average? <5,000 reads?

Yes, we subsampled OTUs from all hosts until 5000 was reached. We have added this information to the Figure legend.

L215- What percentage of OTUs were in >5% of samples?

98% of OTUs were present in <5% of samples. We have added this information to the manuscript text.

Discussion

L470- Are the subnetwork shifts among diets statistically significant?

We thank the reviewer for suggesting this additional analysis. We did find the subnetwork shifts to be statistically significant. The text has been updated to include our hypothesis testing results.

L548- “would expand or contract alpha-diversity”. It would be clearer to say increase or decrease.

We have changed “expand” and “contract” to “increase” and “decrease”, respectively.

L587- “increased microbial yield and fiber digestion”. Sentence is unclear. Do you mean greater microbial richness and ability to digest fiber?

We are referring to yield of microbial biomass that the host then metabolises. We have updated the text to clarify this point.

L602- “This is consistent with a hypothesis that the effects of environmental filtering are limited among Artiodactyla compared to processes selecting for unrelated taxa.” Sentence reads as though processes are deliberately selecting for unrelated taxa.

We have removed the last portion of this sentence, given that it introduced confusion and was not necessary to convey the appropriate message.

Reviewers' Comments:

Reviewer #1:

Remarks to the Author:

This manuscript ("Host diet and evolutionary history explain different aspects of gut microbiome diversity among vertebrate clades") is a revised version of a previously submitted paper.

The authors have satisfactorily responded to most of my comments, including by incorporating new analyses that generated exciting results. I find this manuscript of high quality, and I anticipate that it will be very useful to the field, and interesting to a large audience.

However, I still have issues with the discussion of some of the results and their interpretation in the context of previous literature. First, in the section "Host diet and phylogeny modulate different aspects of gut microbial diversity" (L605), authors mention that host phylogeny associates with bacterial taxonomic groups at low phylogenetic resolutions (OTUs). They later claim that this confirms the results of Nishida and Ochman (L628). It's true, but Sanders et al (Mol Ecol, 2014) were the first to show such a result, later confirmed by Groussin et al (Nat Comm, 2017).

Second, the fact that the authors claim that their results contrast with findings of Groussin et al on the correlation between diet and large bacterial taxonomic groups is interesting (L642). However, I'm not convinced by the hypothesis proposed by the authors to explain this difference, namely that analyzing only captive animals with artificial diets would (surprisingly, because it's not really clear how) have somehow generated significant correlations between diet and large bacterial groups. If captivity and zoo diet would drive this difference, then correlation between diets and beta diversities would change when only looking at wild animals. However, Supp Fig 10 shows that beta diversities associate with diet in the same magnitude as in Fig 2 after excluding captive animals. Overall, I don't think there are results in the main text that support that the diet of zoo animals used in this study has an impact on gut microbiomes when compared to wild counterparts. Maybe the difference lies in the differences in dietary information or in the resolution of dietary distances between the two studies? I'm not asking that the authors solve this issue here, and they can leave it open for future investigations. But this interpretation would make much more sense to me if the dietary data used by the two studies are indeed different.

Title of Supplementary Fig. 9 should be edited ('just' is repeated twice) – "Very similar MRM results obtained if just selecting just one sample per family, which reduces sample size biases towards Mammalia."

Reviewer #2:

Remarks to the Author:

Overall, Youngblut et al's revised manuscript does a thorough job in addressing the major concerns from the initial submission, and the updated manuscript is much improved. There are only a few concerns that should still be addressed. First, we appreciate that <19% of taxa in the dataset have genomes available and so creating a phylogeny for the dataset in the paper is impractical. The manuscript would benefit from describing in the text why timetree was used. Second, the addition of panels B and C to Fig. 1 is helpful. However, the diet detailed label is still not well-explained. For example, is black yes or no? Adding an example of a specific animal's diet to the legend could be useful. "For example, Capra ibex consumes diet types 1,2, and 3 (indicated by yellow), but not diet types 4-8 (black)." Finally, the authors did not address the comment that some samples in Fig.1 appear to be comprised of just 1 phylum; Nyctalus noctula and Crocidura suaveolens (bottom left),

are completely Proteobacteria. It would be good to include an explanation of why this is not driven by sequencing errors or contamination.

REVIEWERS' COMMENTS:

Reviewer #1 (Remarks to the Author):

This manuscript (“Host diet and evolutionary history explain different aspects of gut microbiome diversity among vertebrate clades”) is a revised version of a previously submitted paper.

The authors have satisfactorily responded to most of my comments, including by incorporating new analyses that generated exciting results. I find this manuscript of high quality, and I anticipate that it will be very useful to the field, and interesting to a large audience.

However, I still have issues with the discussion of some of the results and their interpretation in the context of previous literature. First, in the section “Host diet and phylogeny modulate different aspects of gut microbial diversity” (L605), authors mention that host phylogeny associates with bacterial taxonomic groups at low phylogenetic resolutions (OTUs). They later claim that this confirms the results of Nishida and Ochman (L628). It’s true, but Sanders et al (Mol Ecol, 2014) were the first to show such a result, later confirmed by Groussin et al (Nat Comm, 2017).

We agree with the reviewer that we should cite all three relevant studies. We have updated this section of the manuscript to reflect this.

Second, the fact that the authors claim that their results contrast with findings of Groussin et al on the correlation between diet and large bacterial taxonomic groups is interesting (L642). However, I’m not convinced by the hypothesis proposed by the authors to explain this difference, namely that analyzing only captive animals with artificial diets would (surprisingly, because it’s not really clear how) have somehow generated significant correlations between diet and large bacterial groups.

Multiple studies have shown that captivity can substantially alter gut microbiome diversity, even at the coarse taxonomic level of phylum (e.g., those cited in the Introduction). Therefore, we believe that the correlation detected by Groussin et al., was feasibility a result of studying only captive animals.

If captivity and zoo diet would drive this difference, then correlation between diets and beta diversities would change when only looking at wild animals. However, Supp Fig 10 shows that beta diversities associate with diet in the same magnitude as in Fig 2 after excluding captive animals.

This argument assumes that the effect size of captivity is (nearly) equal in all captivity settings, and that the effect size is so large that including/excluding only 20% non-wild animals would significantly alter our findings.

Overall, I don't think there are results in the main text that support that the diet of zoo animals used in this study has an impact on gut microbiomes when compared to wild counterparts. Maybe the difference lies in the differences in dietary information or in the resolution of dietary distances between the two studies? I'm not asking that the authors solve this issue here, and they can leave it open for future investigations. But this interpretation would make much more sense to me if the dietary data used by the two studies are indeed different.

While both our study and Groussin et al., used detailed dietary information, the categories did differ somewhat between the studies (e.g., Groussin et al., used 9 dietary categories, while our study used 8). So we agree with the reviewer that this difference in dietary information may have contributed to the differences observed between the studies. However, it is hard to discern how much this discrepancy contributed to the different findings, in comparison to the differences in captivity status, or other experimental design and analysis methodological differences between these two studies.

Title of Supplementary Fig. 9 should be edited ('just' is repeated twice) – “Very similar MRM results obtained if just selecting just one sample per family, which reduces sample size biases towards Mammalia.”

We thank the reviewer for pointing out this typo. It has been corrected.

Reviewer #2 (Remarks to the Author):

Overall, Youngblut et al's revised manuscript does a thorough job in addressing the major concerns from the initial submission, and the updated manuscript is much improved. There are only a few concerns that should still be addressed. First, we appreciate that <19% of taxa in the dataset have genomes available and so creating a phylogeny for the dataset in the paper is impractical. The manuscript would benefit from describing in the text why timetree was used.

We agree that this information would help clarify our motivations for using a phylogeny from timetree.org. We have updated the “Host phylogeny” sub-section of the Methods to incorporate this information.

Second, the addition of panels B and C to Fig. 1 is helpful. However, the diet detailed label is still not well-explained. For example, is black yes or no? Adding an example of a specific animal's diet to the legend could be useful. “For example, *Capra ibex* consumes diet types 1,2, and 3 (indicated by yellow), but not diet types 4-8 (black).”

We have edited the figure legend to include a more detailed description of the coloring scheme, and we have added an example as suggested.

Finally, the authors did not address the comment that some samples in Fig.1 appear to be comprised of just 1 phylum; *Nyctalus noctula* and *Crocidura suaveolens* (bottom left), are

completely Proteobacteria. It would be good to include an explanation of why this is not driven by sequencing errors or contamination.

While we cannot fully rule out the contribution of sequencing errors or contamination, the dominance of Proteobacteria in these samples has support from past studies. A high relative abundance of Proteobacteria has previously been observed among most Chiroptera species (e.g., Nishida and Ochman 2018, Molecular Ecology). As for Crocidura suaveolens, there are few studies that assess the gut microbiome of Crocidura (or more broadly Soricidae), but LI et al., 2014 Zoological Research did find that Proteobacteria were most prevalent among 12 wild tree shrews. Also, we do not observe Proteobacteria dominating most/all samples in the dataset, suggesting that a general contamination during the NGS library preparation was not the cause. While sequencing errors could lead to some mis-classifications of OTU taxonomy, it is highly unlikely that most/all mis-classifications of OTUs would falsely be Proteobacteria to the exclusion other phyla. We have edited the results to better clarify that other studies do indeed support the dominance of Proteobacteria in certain animal species.